# PLK1 facilitates chromosome biorientation by suppressing centromere disintegration driven by BLM-mediated unwinding and spindle pulling

Owen Addis Jones[1], Ankana Tiwari[1,2], Tomisin Olukoga[1,2], Alex Herbert [1] & Kok-Lung Chan [1]

Centromeres provide a pivotal function for faithful chromosome segregation. They serve as a foundation for the assembly of the kinetochore complex and spindle connection, which is essential for chromosome biorientation. Cells lacking Polo-like kinase 1 (PLK1) activity suffer severe chromosome alignment defects, which is believed primarily due to unstable kinetochore-microtubule attachment. Here, we reveal a previously undescribed mechanism named 'centromere disintegration' that drives chromosome misalignment in PLK1-inactivated cells. We find that PLK1 inhibition does not necessarily compromise metaphase establishment, but instead its maintenance. We demonstrate that this is caused by unlawful unwinding of DNA by BLM helicase at a specific centromere domain underneath kinetochores. Under bipolar spindle pulling, the distorted centromeres are promptly decompacted into DNA threadlike molecules, leading to centromere rupture and whole-chromosome arm splitting. Consequently, chromosome alignment collapses. Our study unveils an unexpected role of PLK1 as a chromosome guardian to maintain centromere integrity for chromosome biorientation.

[1] Genome Damage and Stability Centre, University of Sussex, Brighton BN1 7BG, UK. [2] These authors contributed equally: Ankana Tiwari, Tomisin Olukoga. Correspondence and requests for materials should be addressed to K.-L.C. (email: koklung.chan@sussex.ac.uk)

Chromosome mis-segregation has wide implications in cancer and rare congenital disorders[1]. To achieve faithful chromosome segregation, condensed chromosomes need to be properly aligned prior to disjunction through a mitotic process called chromosome biorientation. This requires a stable connection of spindle microtubules (MTs) emanating from opposite centrosomes to centromeres via the macromolecular complex of kinetochores (KTs)[2]. A single unattached chromosome can activate the spindle assembly checkpoint, inhibiting the anaphase promoting complex/cyclosome (APC/C)[3], and hence blocks anaphase onset[4,5]. This elegant system allows cells to correct possible KT-MT attachment errors and prevent chromosome mis-segregation. During chromosome biorientation, centromeres and KTs are inevitably under constant spindle pulling tension, due to the persistence of sister chromatid cohesion. The centromere architecture is presumably maintained through chromosome condensation, whilst the KT-MT stable attachment requires activity of a key mitotic kinase, Polo-like kinase 1 (PLK1)[6–8]. In early mitosis, PLK1 localises predominantly at KTs and centrosomes. Inactivation of PLK1 has been shown to induce severe chromosome misalignment, which is generally attributed to a failure in building stable KT-MT attachment. However, how PLK1 promotes chromosome biorientation still requires investigation

Once biorientation is achieved on every chromosome, the spindle checkpoint is satisfied. This leads to the activation of APC/C and cleavage of cohesin, allowing the poleward movement of sister chromatids[9]. Interestingly, studies show that despite their separation, sister chromatids can remain intertwined by DNA linkage molecules that manifest as so-called ultrafine DNA bridges (UFBs)[10,11]. Generally, UFBs are thought to be unresolved double-stranded DNA catenanes, especially those that arise at centromeres[12]. However, studies have also shown that incomplete replication intermediates and homologous recombination (HR) structures can give rise to UFB structures[13–16]. Regardless of their origins, UFBs are recognised by a UFB-binding complex comprising of PICH (Plk1-interacting checkpoint helicase) translocase, BLM (Bloom's syndrome) helicase and its interacting factors, including TOP3A and TOP2[10,11,17–19]. However, the precise molecular mechanism of UFB resolution is not yet fully understood

Chromosome biorientation not only plays a critical role to ensure equal chromosome segregation, but also facilitates the regulation of the spindle checkpoint and mitotic progression. Many studies have shown that PLK1 is essential for chromosome biorientation; however, the underlying mechanism(s) is still not fully clear. In the current study, unexpectedly, we find that PLK1 in fact can protect centromere integrity for chromosome biorientation maintenance. We demonstrate that in the absence of PLK1, the UFB-binding complex aberrantly targets and unwinds centromeres, leading to their rupture in concerted action with bipolar spindle pulling. As a consequence, cells lose centromere integrity and fail to maintain metaphase alignment. Therefore, our study provides an alternative mechanism of chromosome misalignment in PLK1-defective cells. Importantly, it also reveals a previously undescribed pathway of centromere protection during mitosis

## Results

**PLK1 inactivation leads to collapse of metaphase alignment.** It is well-documented that cells cannot achieve proper chromosome alignment without PLK1 activity[6]. Consistent with this, inhibition of PLK1 using a well-characterised small molecule inhibitor, BI2536 (IC$_{50}$ = 0.83 nM)[20], induced severe chromosome misalignment in hTERT-immortalised human RPE1 cells. The BI2536-induced mitotic arrest manifested in a way similar to treatments of the spindle poison (nocodazole) and kinesin inhibitor (Monastrol) (Supplementary Fig. 1a). However, live-cell time-lapse microscopy on pre-synchronised RPE1 cells revealed that, unlike nocodazole and monastrol treatments, BI2536 did not fully prevent chromosome congression (Fig. 1a and Supplementary Fig. 1b). Nearly 80% of BI2536-treated RPE1 cells managed to align their chromosomes in the metaphase plane, but shortly after, succumbed to a loss of maintenance; namely chromosomes drifting away from the equator and scattering into a 'Fig-8' or 'polo'[21]-like pattern (Fig. 1a–c, Supplementary Fig. 1c and Supplementary Movie 1). We referred to this phenomenon as 'metaphase collapse'. In contrast, cells treated with the APC/C inhibitor, ProTAME, remained arrested at metaphase for extended periods (Fig. 1c)

**Formation of centromeric DNA linkages between chromosomes.** Strikingly, in the metaphase-collapse cell population, we observed a threadlike structure that was decorated by the PLK1 protein (Fig. 2a; arrows). It was not present in DMSO-treated pre-anaphase cells (i.e., prometaphase and metaphase) (Fig. 2a). As the threadlike structure was reminiscent of anaphase UFBs[10,11], we investigated whether they were DNA molecules; or a mis-localisation of PLK1 to cytoskeleton structures. Immunofluorescence co-staining showed that PICH translocase, a well-known UFB marker, was present along the PLK1-coated threads induced by BI2536 (Fig. 2b and Supplementary Fig. 1d). Furthermore, other known UFB-associated factors including BLM helicase and replication protein A (RPA) were also present (Fig. 2c). It is worth noting that RPA decorates the threads without necessarily following the PICH/BLM signals, and it can also be found on regions where no or weak PICH/BLM signals were detected (Fig. 2c; arrows). Similar results were obtained by using different antibodies against BLM and different subunits of RPA (Fig. 2d). This localisation pattern is similar to recent reports showing the binding of RPA to stretched DNA molecules, or DNA bridges, is not always coupled with the PICH/BLM complex[12,22]. Therefore, the RPA association likely represents the presence of single-stranded DNA. To validate the immunofluorescence staining results, we examined Bloom's syndrome fibroblast cells stably expressing a GFP-tagged BLM, and RPE1 cells expressing a GFP-tagged PLK1. We found both GFP-tagged proteins were also present along the thread molecules induced by BI2536 (Supplementary Fig. 1e, f; arrows). In addition to this, we also found that PLK1 indeed associated with UFBs in anaphase cells (Supplementary Fig. 1g; arrows). Together, these data suggest that the BI2536-induced thread molecules are highly likely a form of DNA structure; and possibly composed of both single-stranded and double-stranded DNA. As predicted, the threadlike structures did not co-localise with mitotic MTs (Supplementary Fig. 1h)

Next, we investigated the origin of the DNA threads. We found that all of the DNA threads analysed linked to centromeric regions, either through one or both of their termini (Fig. 2e). In some optical sections, it was apparent that two separating centromeres were inter-connected by a DNA thread (Fig. 2e; arrows). Since PLK1 inhibition prevents anaphase onset, these DNA threads cannot be explained as the centromeric UFBs coming from disjoined sister chromatids. However, another possibility is that PLK1 inhibition might induce precocious sister chromatids separation; an effect similar to Shugosin 1 (SGO1) depletion[23,24] (Supplementary Fig. 2a), which exposes UFBs before anaphase[11] (Supplementary Fig. 2b; arrows). This proposal, however, is very unlikely because PLK1 has been shown to be required for the release of arm cohesin; its

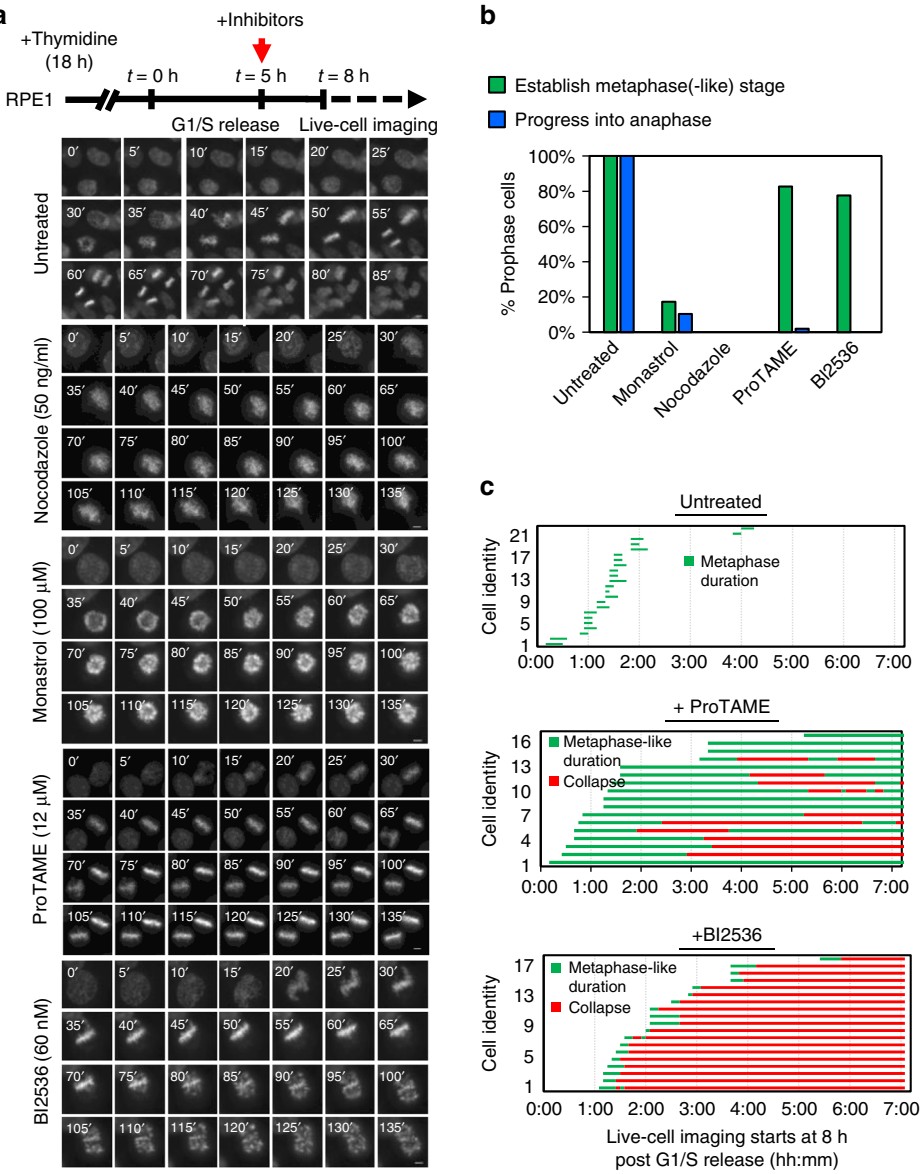

**Fig. 1** PLK1 inactivation causes metaphase collapse. **a** Experimental outline and time-lapse imaging examples showing the mitotic progression of RPE1 cells under the indicated drug treatment. RPE1 cells were arrested in G1/S by a single thymidine block. Five hour post-G1/S-release, the indicated inhibitors were added for another 3 h, followed by live-cell microscopy (started at $t = 8$ h post-G1/S release). Time-lapse imaging was carried out with a 5-min interval. **b** Percentages of prophase cells establishing metaphase(-like) alignment and progressing into anaphase under the indicated treatment. A total of 41, 29, 21, 52 and 58 prophase cells were analysed in untreated, monastrol, nocodazole, ProTAME- and BI2536-treated conditions. **c** The durations of metaphase (-like) and collapse stages (h:min) of RPE1 cells under the indicated treatment. Note: only cells that established a metaphase(-like) stage were measured. 23, 17 and 19 cells were analysed in untreated, ProTAME- and BI2536-treated conditions. DNA was stained by SiR-DNA. Scale bars = 5 μm

inactivation in fact blocks premature loss of cohesion[25–27]. Consistent with these studies, we confirmed that BI2536 did not induce premature separation of sister chromatids in RPE1 cells (Supplementary Fig. 2c). More importantly, by co-staining with topoisomerase 2alpha (TOP2A), a chromatid-axis marker, we visualised that cohesed chromosomes were linked by the DNA threads, induced by PLK1 inhibition (Fig. 2f; arrows). Therefore, the absence of PLK1 activity leads to the formation of a DNA linkage structure that strikingly connects centromeres between cohesed chromosomes. Because of the concomitant occurrence of metaphase collapse and centromere DNA linkages, we speculate that the failure of chromosome alignment in PLK1-inactivated cells may not be merely attributed to unstable KT-MT attachments as previously thought.

**Mitotic loss of PLK1 causes centromeric DNA linkages**. The DNA linkages induced by BI2536 arise predominantly at centromeres—a genomic region composed of highly repetitive sequences. We sought to test if they might be caused by potential disturbance of DNA replication (or HR) during the course of BI2536 treatment. We used EdU labelling to distinguish between cells that were in an ongoing, or post DNA replication stage, whilst under BI2536 treatment (Supplementary Fig. 3a). If centromere DNA threads are a by-product of abnormal DNA replication, we expected to observe their formation only in the EdU-positive, but not negative, mitotic population. Contrary to this hypothesis, we found that the majority of EdU-negative mitotic cells (69 ± 4%), which were presumably in G2/M while BI2536 was applied, remained positive for DNA thread formation (Supplementary

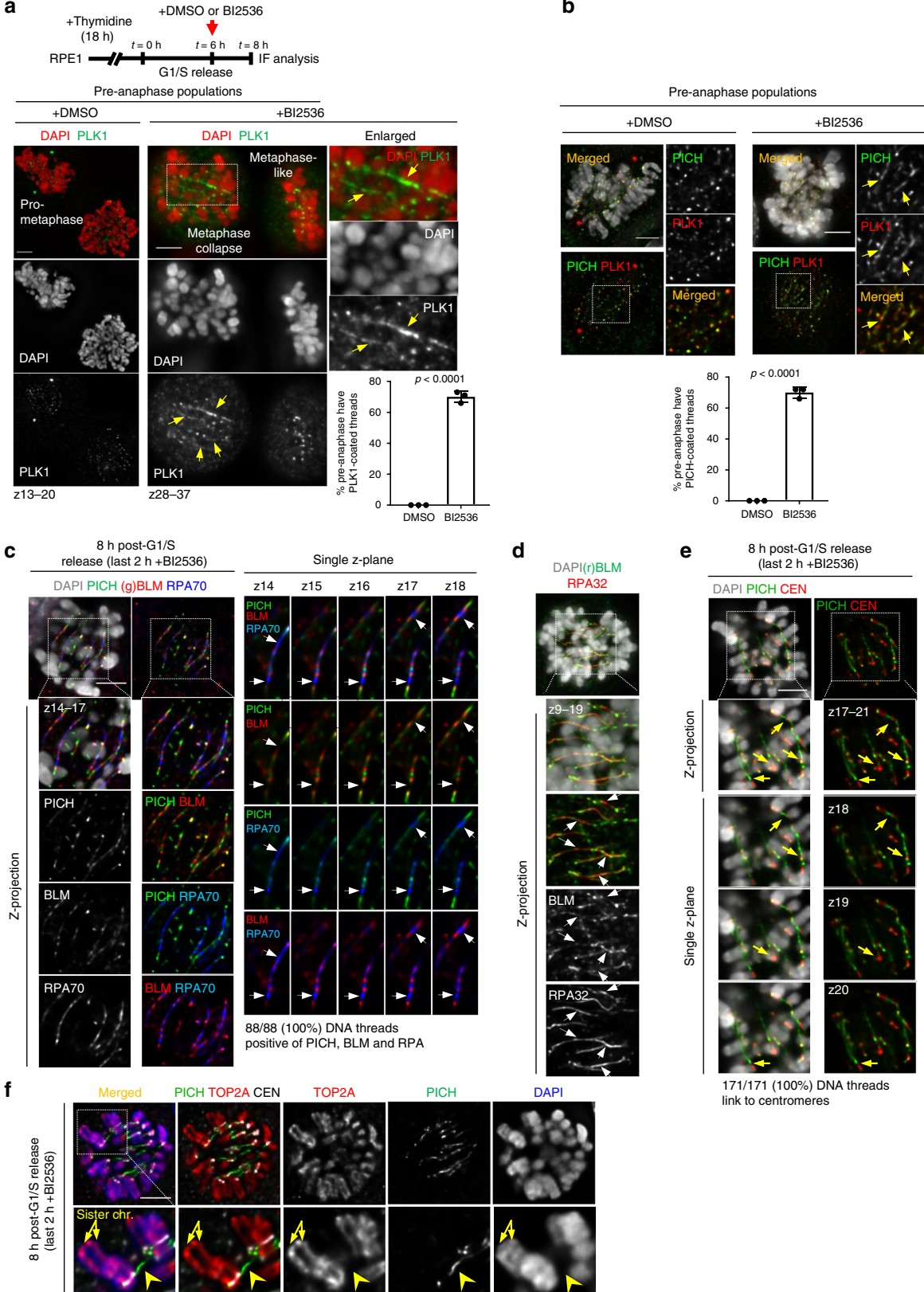

Fig. 3b). Moreover, inhibition of PLK1 in early mitotic RPE1 cells obtained through a release from RO3306-induced G2 arrest also induced centromere DNA threads (Supplementary Fig. 3c). In addition, treating asynchronous RPE1 cells with BI2536 for 1 h also caused centromere DNA thread formation, albeit with a lower frequency (Supplementary Fig. 3d). These data indicate that the formation of centromere DNA linkages likely results from a loss of M-phase specific function of PLK1. However, thymidine pre-treatment and/or synchronistic mitotic entry may enhance the phenotype appearance

**Fig. 2** Formation of centromere DNA linkages after PLK1 inactivation. **a** Experimental outline (top) and representative images showing the immunofluorescent staining of PLK1 in DMSO- and BI2536-treated RPE1 pre-anaphase cells (prometaphase and metaphase). BI2536 induced the formation of a PLK1-decorated threadlike structure (arrows) in the 'metaphase collapse' cell. Enlarged region is shown at right. Quantification (bottom) showing the percentage of cells positive for PLK1-coated threads (mean ± S.D. is shown; $n = 3$ independent experiments analysing 75 and 224 pre-anaphase cells in DMSO- and BI2536-treated conditions, respectively). **b** The experimental setup is same as in (**a**). PICH co-localises with PLK1 on the threadlike structures induced by BI2536. Quantification (below) showing the percentage of cells positive for PICH-coated threads (mean ± S.D. is shown; $n = 3$ independent experiments analysing 120 and 308 pre-anaphase cells in DMSO and BI2356 treated conditions, respectively). **c** Representative images showing the association of the UFB-binding complex (PICH, BLM and RPA70) along the threadlike molecules induced by BI2536. Enlarged regions of both z-projected (below) and single z-planes (right) are shown. Eighty-eight thread structures were examined and all were positive for PICH, BLM and RPA70 staining. Arrows showing regions where the RPA70 staining is strong but with no or weak PICH/BLM signals. Note: BLM was stained by a goat antibody (C-18). **d** RPA32 localises on the thread regions where BLM signal is weak (arrows). Note: BLM was stained by a rabbit antibody (ab2179). **e** DNA threads link between centromeres (arrows). Enlarged regions of z-projection and single z-planes are shown below. Note: all DNA threads examined (171/171 in 10 cells; 100%) are positive for centromere linkages at either one or both of their termini. **f** Representative images showing cohesed sister chromatids as labelled by TOP2A (arrows) are linked by DNA threads at their centromeres (arrowhead). The enlarged region is shown below. DNA was stained by DAPI. Scale bars = 5 μm

Using the same treatment protocol, we also found that BI2536 induced centromere DNA threads in all other examined cell types, though with different frequencies (Supplementary Fig. 4a). They included 1BR3 primary fibroblasts (31%), 82-6 hTERT-immortal fibroblasts (24%), HCT116 colon (69%) and HeLa cervical cancer cells (21%). Since the DNA thread formation occurs following metaphase collapse, the different frequencies between cell lines, (e.g., RPE1 vs. HeLa), may relate to their ability to establish metaphase. In agreement with this, time-lapse microscopy revealed that, as compared to RPE1, HeLa cells poorly progressed into a metaphase(-like) stage under BI2536 treatment (Supplementary Fig. 4b). These data are consistent with other studies[27,28], and may indicate that the formation of bipolar spindle attachment in HeLa cells is more sensitive to the loss of PLK1 activity

**PLK1 inactivation induces whole-chromosome arm splitting**. As shown above, centromere DNA threads cannot be described as originating from the DNA entanglements between sister chromatids/centromeres. We thus investigated other possible cause(s). PICH translocase binds with a high affinity to DNA molecules under tension[29]. This could indicate that the DNA threads may be a form of abnormally stretched centromeric chromatin. Interestingly, we detected activation of DNA damage responses at (peri)centromeric regions, as labelled by γH2AX staining, following BI2536 treatment (Supplementary Fig. 5a). The damage response was mainly observed in metaphase collapse populations rather than in early mitotic cells (e.g., prophase/early prometaphase) (Supplementary Fig. 5a). As expected, γH2AX was mostly not detected at (peri)centromeric regions in DMSO-treated mitotic cells (Supplementary Fig. 5b).

To determine if this was caused by chromatin damage, we examined mitotic chromosome spreads (Supplementary Fig. 6a). In control RPE1 cells (DMSO- and nocodazole-treated), their chromosomes displayed normal configurations and their average numbers were very close to 46 (diploid) (Supplementary Fig. 6b, c). In contrast, chromosomes of BI2536-treated cells exhibited a shorter and more compact structure, but strikingly, their chromosome numbers increased to an average of 59 (Supplementary Fig. 6b, c). This increment cannot be explained by chromosome mis-segregation, because PLK1 inactivation blocks anaphase onset. Thus, a plausible explanation is chromosome fragmentation. Moreover, we found that the increase in chromosome numbers in BI2536-treated cells was suppressed by co-treatment with nocodazole (Supplementary Fig. 6b, c), implying a spindle (or tension)-dependent process. Furthermore, centromere-telomere fluorescence in situ hybridisation (ctFISH) analysis confirmed that the mitotic chromosomes in BI2536-

treated cells were indeed broken (Fig. 3a and Supplementary Fig. 7). Notably, the broken chromatin largely resembled telocentric chromosomes; namely the centromere residing at one end of the chromatin, but lacking the telomere signals (Fig. 3a, middle panels—asterisks and Supplementary Fig. 7a, middle panels). This pattern suggests that the breakage occurs either at, or very close to the centromere. Supporting this, we also observed partial centromere splitting (Fig. 3a, middle panels—arrowheads and Supplementary Fig. 7a), and occasionally, saw a CEN DNA thread linking two separating broken chromosome arms (Fig. 3a, middle panels—connecting arrow and Supplementary Fig. 7a). Nocodazole treatment again suppressed chromosome arm breakages, but seemed to have a lesser effect on the partial splitting of centromeres (Fig. 3b). Together, these results demonstrate that the loss of PLK1 activity induces centromere rupture in a spindle-dependent manner. In agreement to this, we found that nearly all the broken chromatin (99.6%) retained centromere sequences (Fig. 3c and Supplementary Fig. 7b), indicating that most, if not all of the breakages, occur within the core centromere.

Our results from both cytological and cytogenetic analyses suggest that the centromere DNA threads induced by PLK1 inhibition are highly likely caused by abnormal stretching of the core centromere chromatin by the spindle pulling forces. As predicted, nocodazole suppressed both centromere splitting and DNA thread formation (Fig. 3b and Supplementary Fig. 8). Given that centromeric DNA threads arise mostly after metaphase establishment, we postulated that rather than by spindle-dependent chromosome movement, they are likely mediated by the tension exerting across the centromeres due to 'bipolar' spindle attachment. We thus used Monastrol, the Eg5 inhibitor, to prevent bipolar spindle establishment while keeping MT attachment[30]. As predicted, 'monopolar' spindle attachment is not sufficient to induce centromere DNA threads (Supplementary Fig. 8). Therefore, centromere splitting requires bipolar spindle pulling forces.

To our knowledge, this striking phenomenon of spindle-mediated centromere rupture has never been described; we thus termed this 'centromere dislocation'. Using multi-colour FISH (mFISH) analysis, we further validated that PLK1 inactivation can cause whole-chromosome arm separation (Fig. 3d). In some cases, the separated whole-arms were located in close vicinity (Fig. 3d; e.g., chromosomes 7p–7q, 12p–12q and 17p–17q), which may imply a residual physical connection, presumably through the ultrafine centromeric DNA threads. In addition, centromere dislocation tended to occur more frequently on longer chromosomes (Fig. 3d, inset). Collectively, our data show that, in the absence of PLK1 activity, centromere chromatin fails to withstand

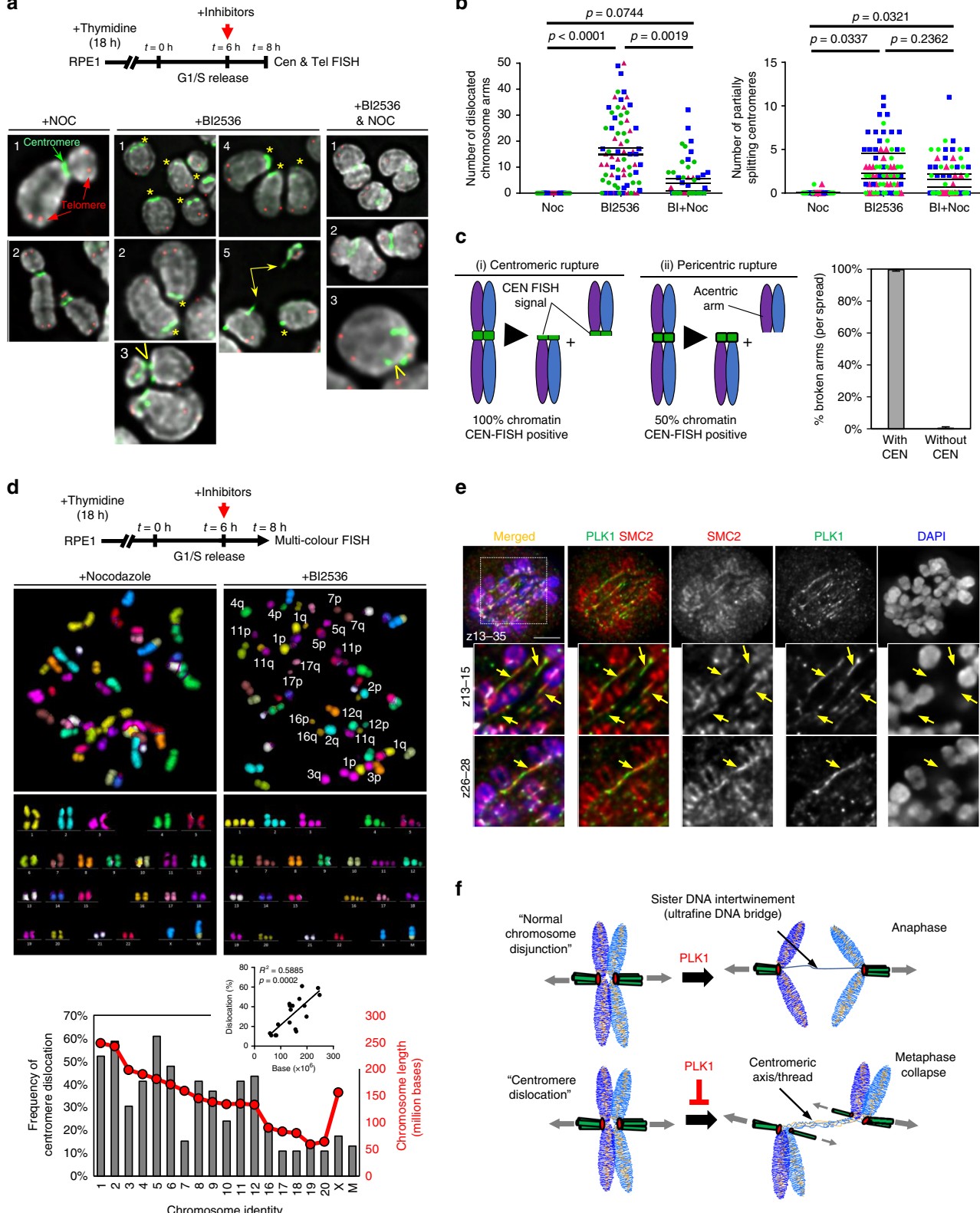

bipolar spindle tensions and the core axis is transformed into an ultrafine DNA threadlike structure. Indeed, we were able to detect condensin, a chromosome axial element, associating along the stretched DNA threads (Fig. 3e; arrows). The disintegration of centromeres therefore causes whole-chromosome arm splitting and explains why cells simultaneously lose their metaphase alignment (Fig. 3f).

**PLK1 kinase activity suppresses centromere disintegration**. Thus far, most of the experiments were carried out using the PLK1 inhibitor, BI2536. To rule out potential off-target effects, such as inhibition to other PLK members[20], we employed an engineered RPE1 cell line in which the endogenous wild-type (WT) PLK1 has been replaced with an analogue-sensitive allele, PLK1as. The catalytic cavity of the PLK1as protein has been

**Fig. 3** PLK1 inactivation induces spindle-dependent centromere dislocation. **a** Experimental outline (top) and representative deconvolved images showing chromosomes isolated from RPE1 cells under the indicated treatment. Chromosomes were hybridised with FISH DNA probes against centromeres (green) and telomeres (red). Left panels: examples of normal chromosome configuration (+nocodazole). Middle panels: examples of BI2536-induced 'centromere dislocations' (asterisks; 1–5), 'partial centromere splitting' (arrowhead; 3), and a centromere DNA thread linking two separate chromosome arms (connecting arrow; 5). Right panels: chromosomes with 'partial centromere splitting' (arrowhead) after BI2536 and nocodazole co-treatment. Note: also see Supplementary Fig. 7 for the whole-chromosome spread images. **b** Quantification of 'chromosome arm dislocations' (left) and 'partially centromere splitting' (right) under the indicated inhibitor treatment ($n = 3$ independent experiments analysing 75 spreads in each condition; the means of each experiment are shown). **c** A diagram depicting the outcomes of chromosome breakage within or outside centromeres. (i) Breakage at centromeres generates both broken arms (100%) positive for CEN FISH signal; (ii) breakage at pericentric or arm regions generates one of the broken arms (50%) positive for a CEN FISH signal. Quantification (right) of the examined broken chromosome arms with or without centromere FISH signal at their termini (524 broken chromosome arms were scored from 11 separate chromosome spreads showing the highest centromere dislocation frequency). **d** Experimental outline (top) and mFISH karyotyping of RPE1 cells. BI2536 induced chromosome 'p'- and 'q'-arm separation. Note: there is a marker 'M' chromosome with a translocation of chromosome X and 10 in RPE1 cells. Bar graph (bottom) showing the frequency of 'centromere dislocations' among individual chromosomes. Inset graph showing the positive correlation between chromosome length and 'centromere dislocation' frequency (23 spreads were analysed). Note: acrocentric chromosomes were not determined and the length of the 'marker' chromosome is unknown. **e** Condensin (SMC2) is detected on some PLK1-associated DNA threads (arrows). Scale bars = 5 μm. **f** A model of centromere dislocation induced by PLK1 inactivation in a spindle-dependent manner. Spindle-mediated tension causes decompaction of centromere axis, the formation of centromere DNA threads and whole-chromosome arm separation

modified such that it no longer binds to BI2536; instead only to the unrelated ATP-analogue, 3-MB-PP1[31] (Supplementary Fig. 9a). As predicted, BI2536 failed to induce metaphase collapse and centromere DNA thread formation in the engineered PLK1as cells. Importantly, these mitotic defects were recapitulated by using 3-MB-PP1 analogue (Supplementary Fig. 9a–d). In addition, depletion of the PLK1 protein in RPE1 cells by RNA interference (RNAi) also induced centromere DNA threads and dislocations (Supplementary Fig. 9e, f), which further rules out the potential dominant effect as a result of trapping an inactive form of PLK1 onto chromatin by the small molecule inhibitors. Therefore, PLK1 kinase activity per se is essential to suppress centromere disintegration.

**Aberrant association of UFB-binding factors to KTs**. The failure of centromeres to withstand bipolar spindle pulling in the absence of PLK1 function might indicate that centromere chromatin structure is impaired. We thus analysed the centromeres in the BI2536-treated RPE1 cells before metaphase collapse occurs. We found that there was a progressive formation of RPA foci at or near KTs; from early prometaphase to metaphase(-like) stages (Fig. 4a). This was also sensitive to nocodazole treatment (Fig. 4a). In control, we rarely detected RPA foci at centromeres in normal metaphase cells (Fig. 4b). More interestingly, we also found increased accumulations of BLM and PICH foci at or near the KTs in the metaphase(-like) cells, again only after BI2536 treatments (Fig. 4c and Supplementary Fig. 10a). Occasionally, PICH was found at the inner centromeres of untreated cells (Supplementary Fig. 10a; yellow arrows), perhaps reflecting the unresolved DNA entanglements between sister centromeres as proposed previously[10,11,13,18,32]. Earlier studies have reported that (phospho)-RPA and BLM foci are observed at centromeres of cytospun chromosomes[33,34]. However, under our experimental conditions, both RPA and BLM foci were rarely detected at centromeres in normal intact mitotic cells (Fig. 4b, c). In contrast, PICH foci were consistently visualised at KTs in normal mitotic cells[10] (Supplementary Fig. 10a). To confirm that PLK1 inactivation also enhances PICH loading, we performed quantitative imaging analysis on co-cultured RPE1 cells, using a mixture of cells expressing either a WT PLK1 or GFP-tagged PLK1as protein. This allowed us to directly compare the relative amount of PICH at KTs. Under BI2536 treatment, there was a marked increase in both intensity and number of PICH foci at KTs in WT PLK1, but not in the GFP-PLK1as cells (Supplementary Fig. 10b–d). Conversely, 3-MB-PP1 induced PICH accumulation

at KTs in the GFP-PLK1as cells (Supplementary Fig. 10b–d). As expected, PICH, BLM and RPA foci were mostly co-localised at KTs (Supplementary Fig. 10e; arrows). Therefore, the UFB-binding complex is aberrantly recruited to KT regions when PLK1 function is compromised. Since BLM and PICH possess activities of DNA unwinding and of DNA displacement, respectively[29,35], this led us to speculate that the increase in centromeric RPA foci formation may be due to illegitimate DNA unwinding.

**Centromere distortion underneath KTs**. The localisation of PICH to KTs is independent of PLK1[18,28] (Supplementary Fig. 9e). Our data show that inactivating PLK1 even increases the binding of PICH, BLM and RPA to KTs. Whether the complex actually targets the centromere chromatin, or is aberrantly enriched at KTs is unclear. To address this, we employed high-resolution microscopy to precisely locate the complex within the territory of centromeres (Supplementary Fig. 11a). We found that PICH localised in centromeres at a position ~160 nm away from the outer KT component, as marked by NUF2 (Supplementary Fig. 11b, d). In a control measurement, the inner KT component, CENPA, was mapped ~100 nm inwards from NUF2 in metaphase cells (Supplementary Fig. 11c, d). The CENPA-NUF2 distance was reduced (~80 nm) in anaphase cells (Supplementary Fig. 11c, d), probably due to a reduction of intra-KT tension following sister chromatids cohesion loss[36,37]. This inward position of PICH suggests that it likely locates at centromere chromatin. Further co-staining of PICH and CENPA confirmed that PICH resides at a centromeric domain ~100 nm beneath CENPA (Fig. 4d and Supplementary Fig. 11d). Likewise, both BLM and RPA foci, were mapped underneath CENPA, with distances of ~120 and ~150 nm, respectively (Fig. 4d and Supplementary Fig. 11d). All of these proteins displayed mirror localisation patterns, reflecting a typical symmetry of sister centromere-KT organisation. We referred to this specific centromere site as 'kinetochore-chromatin' or 'K-chromatin' (Fig. 4e).

The K-chromatin localisation finding is consistent with our notion that centromeric chromatin is probably targeted by the UFB-binding complex after PLK1 inhibition. If the increased formation of RPA foci at K-chromatin reflects aberrant DNA unwinding, this may weaken centromere rigidity to counteract spindle pulling forces. Notably, we observed detachments of KT complex in a small population of centromeres (<4%) in metaphase(-like) cells prior to collapse. Intriguingly, some of the KTs remained connected by a short thread, as labelled by

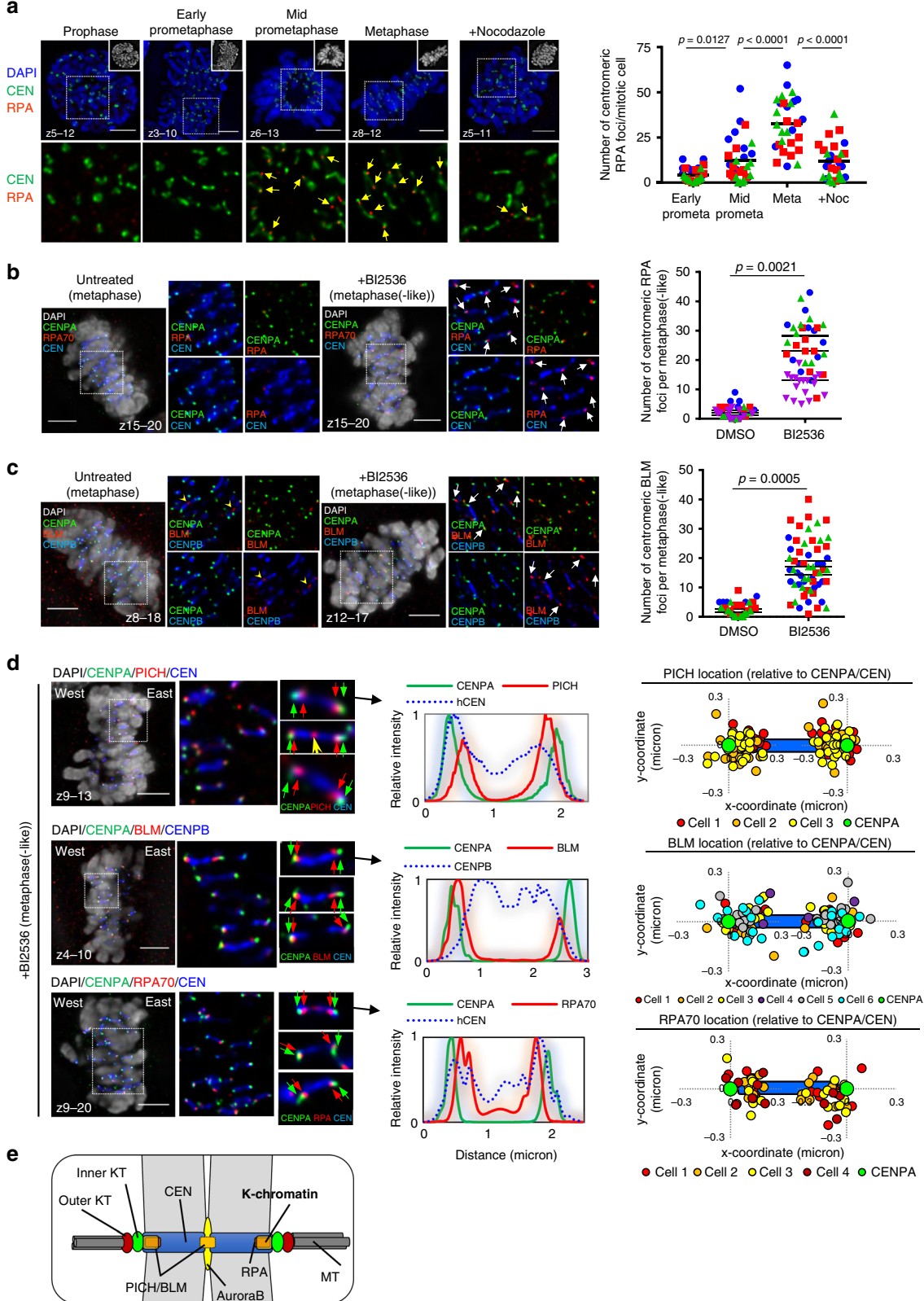

PICH or RPA staining (Fig. 5a, b; arrows). Centromere staining was sometimes evident along the short thread (Fig. 5a; arrows), implying a protrusion of centromeric DNA. We postulated that this might be the early sign of centromere disintegration. Further analysis of the centromere-KT integrity revealed that there was a large percentage of centromeres losing one of the two sister KTs after metaphase collapse (Fig. 5b–d). The side of the centromere where a KT was missing was concomitant with the formation of DNA thread linkages (Fig. 5c; connecting arrows). Therefore, there are apparent alterations on the centromere-KT configuration prior to and during centromere disintegration.

**Fig. 4** PLK1 inactivation increases PICH, BLM and RPA foci at K-chromatin. **a** Increased RPA foci formation at centromeres during mitotic progression after PLK1 inhibition (from prophase to metaphase-like stage). Right: Quantification of the numbers of RPA foci at centromeres in the indicated mitotic stages and treatments ($n = 3$ independent experiments analysing a total of 30 cells in each stage of early, mid prometaphase and metaphase; and of 29 cells in the nocodazole-treated condition; average mean is shown). **b** BI2536 increased the formation of centromeric RPA foci in precollapsed metaphase-like cells. Representative images comparing RPA foci at centromeres in DMSO- (left) and BI2536-treated (right) metaphase-like cells. Enlarged images of the selected regions are shown at right. Arrows indicate centromeric RPA foci. Quantification of RPA foci number at centromeres of metaphase (DMSO), and metaphase(-like) (BI2536) cells ($n = 3$ independent experiments analysing 47 cells per condition). **c** Same as (**b**), but stained with BLM ($n = 3$ independent experiments analysing 59 and 60 cells in DMSO- and BI2536-treated conditions; means of each experiment are shown). **d** Mapping the locations of PICH/ BLM/RPA complex at centromeres. Representative images showing the relative locations of PICH (top), BLM (middle) and RPA (bottom) at the centromeres, comparing to the inner kinetochore marker, CENPA. Profile plots of signal intensity accompanies each example. Right: graphs showing the relative position of each protein at both sides of the centromere. **e** A model depicts the localisation of the PICH/BLM/RPA complex at a specific domain of centromeres, named kinetochore-chromatin/K-chromatin. Note: all RPE1 cells analysed were pre-synchronised at G1/S by a single thymidine block. Drugs were added at 6 h post-release. After 2 h treatment, cells were subject to immunofluorescence staining. Scale bars = 5 μm

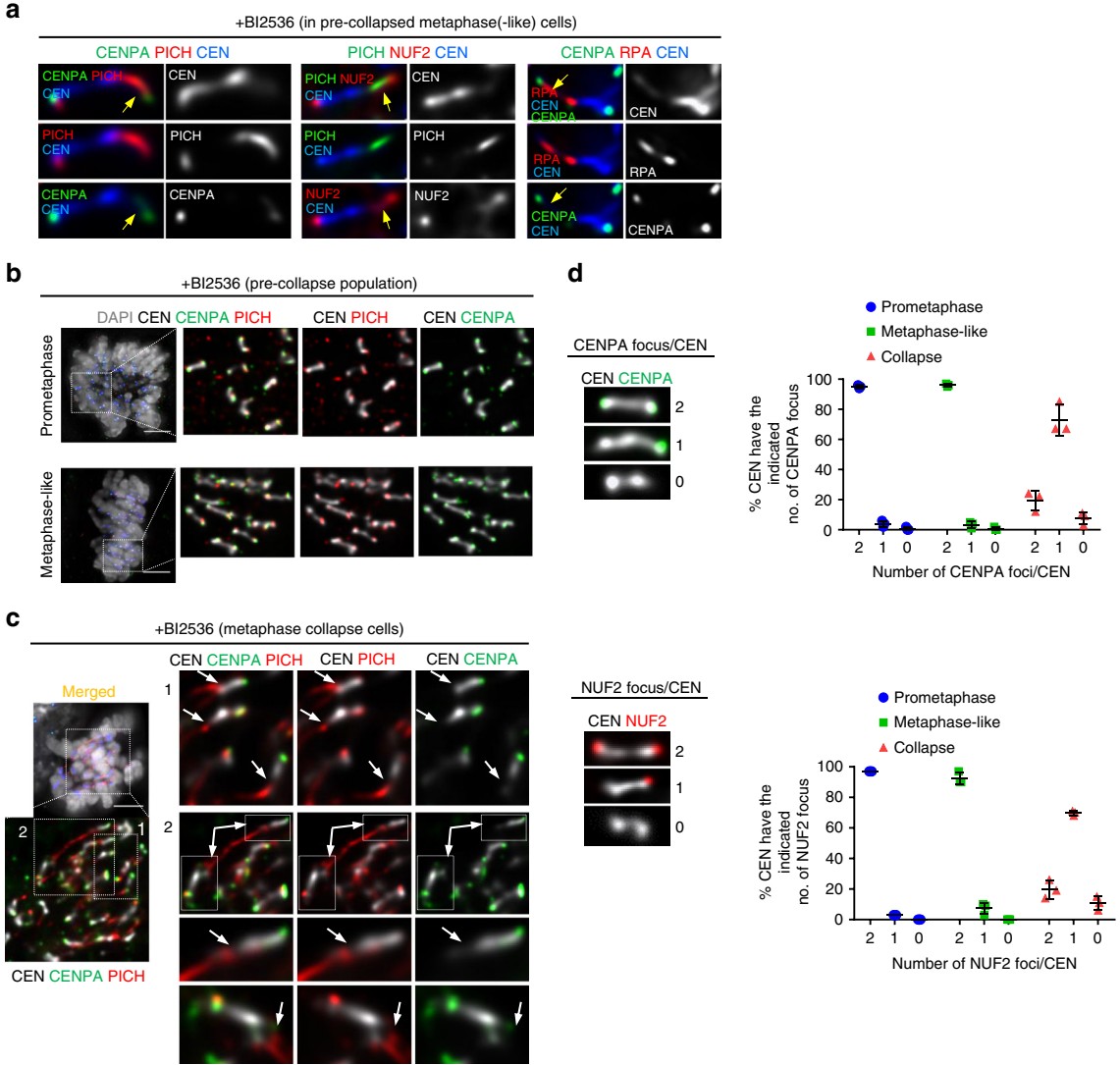

**Fig. 5** Loss of kinetochore attachment at centromeres after dislocation. **a** Representative images showing the kinetochore complex detaches from the core centromere, whilst remaining connected by PICH- or RPA-coated DNA threads (arrows) in BI2536-treated metaphase-like RPE1 cells. Inner and outer kinetochores were labelled by CENPA and NUF2, respectively. **b** Examples showing the majority of centromeres retain two kinetochores in pre-collapsed mitotic populations (prometaphase and metaphase-like cells) after BI2536 treatment. **c** Representative image showing the metaphase-collapse cells losing kinetochore complex at one side of the centromere. Note: the side without the kinetochore is concomitant with the formation of PICH-associated DNA linkages (arrows). Enlarged images (1 and 2) highlight the loss of CENPA signal at regions of where PICH-decorated DNA linkages form (arrows).
**d** Quantification of the numbers of CENPA- and NUF2-labelled kinetochores at centromeres in prometaphase, metaphase-like and collapse stages after BI2536 treatment ($n = 3$ independent experiments analysing a total of 3020 CENPA-labelled centromeres and 3625 NUF2-labelled centromeres; mean ± S. D. is shown). Note: all RPE1 cells analysed were pre-synchronised at G1/S by single thymidine block. Drugs were added at 6 h post-release. After 2 h treatment, cells were subject to immunofluorescence staining. Scale bars = 5 μm

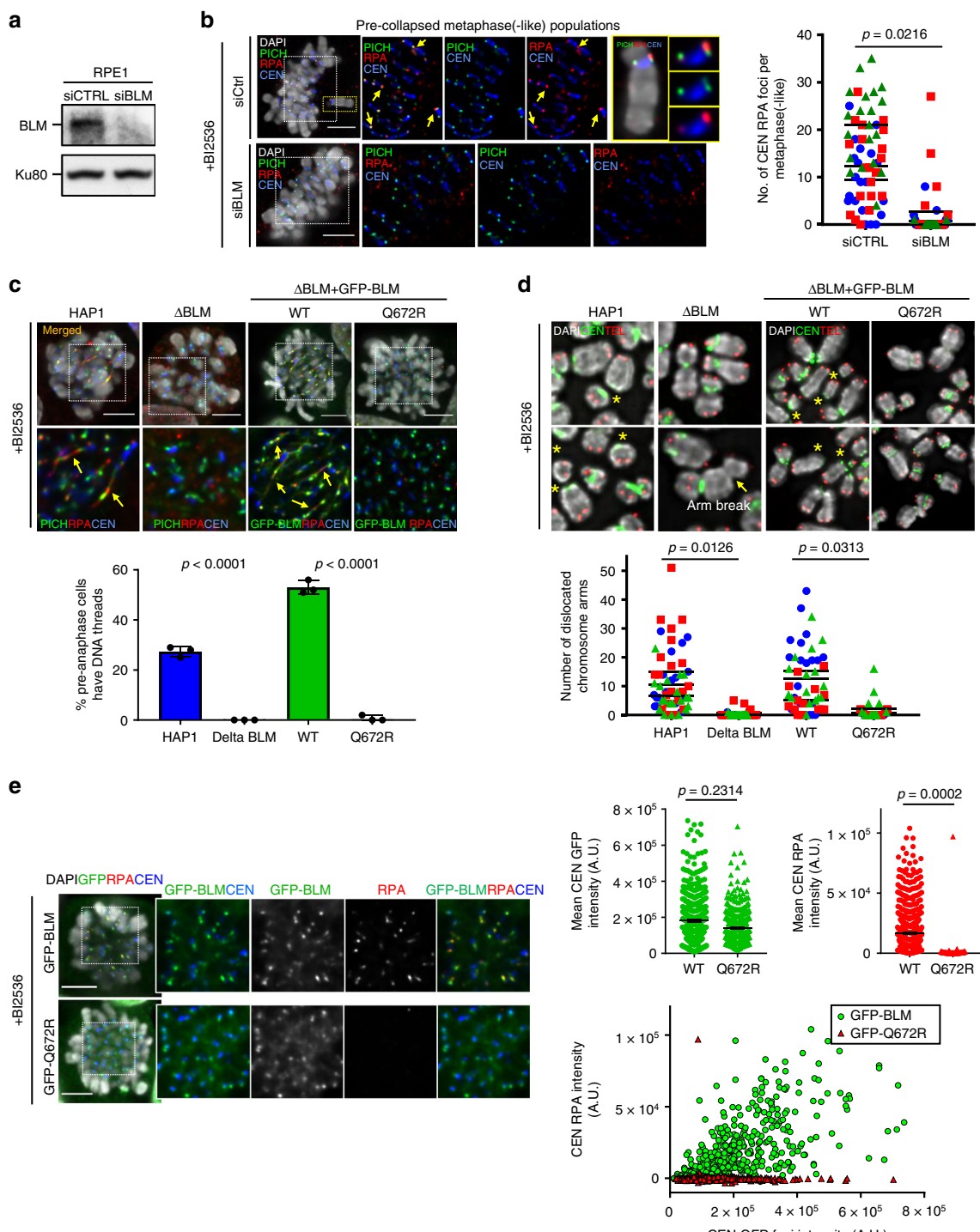

**BLM helicase activity mediates centromere disintegration.** Next, we examined whether centromere disintegration is mediated by BLM and PICH. We knocked down BLM by RNAi before BI2536 treatment (Fig. 6a). Silencing BLM for longer than 48 h in RPE1 cells reduced the efficiency of thymidine release, therefore we treated cells with siBLM oligos for only 24 h prior to G1/S release (Supplementary Fig. 12a). Despite partial depletion (Supplementary Fig. 12b–d), we found that first, BLM knockdown significantly reduced the formation of RPA foci at K-chromatin induced by BI2536 (Fig. 6b); second, it also diminished both centromere DNA thread formation and centromere dislocation (Supplementary Fig. 12e, f). BLM depletion did not impair PICH centromeric localisation (Supplementary Fig. 12g),

suggesting that, without BLM, PICH alone is not sufficient to drive centromere disintegration. To confirm the specificity of BLM knockdown, we performed our analyses on HAP1 cells in which the endogenous BLM was knocked out by CRISPR genome editing[38] (Supplementary Fig. 13a, b). Consistently, BLM knockout abolished centromere DNA thread formation and centromere dislocations (Fig. 6c, d). Though, occasionally, chromatid breaks were observed in ΔBLM HAP1 cells, the breakpoint was not at the centromere (Fig. 6d, arrow). Therefore, in addition to the bipolar spindle pulling forces, BLM is a key driver of centromere disintegration after the loss of PLK1 activity.

Centromere disintegration might be initiated through unlawful DNA unwinding by BLM. Thus, we determined if BLM's helicase

**Fig. 6** BLM helicase activity triggers centromere disintegration. **a** Western blot showing BLM depletion after RNAi treatment in RPE1 cells. Ku80 is used as a loading control. **b** BLM depletion reduced centromeric RPA foci formation induced by BI2536 in pre-collapsed metaphase(-like) cells. Representative images showing the loss of RPA, but not PICH foci, at centromeres in siBLM cells. Right: quantification of centromeric RPA foci in metaphase(-like) cells ($n = 3$ independent experiments analysing 60 cells per condition; mean ± S.D. is shown). **c** Representative images and quantification of DNA thread formation in wild-type (HAP1) cells, BLM knockout cells (ΔBLM), and ΔBLM HAP1 cells complemented with a wild-type GFP-BLM (WT) and a BLM-helicase mutant (Q672R) protein under BI2536 treatment ($n = 3$ independent experiments analysing a total of 291, 218, 204 and 184 cells in HAP1, ΔBLM, WT and Q672R cell lines; mean ± S.D. is shown). **d** Representative chromosome images of 'centromere dislocations' (yellow asterisks) in the indicated HAP1 cells shown in (**c**). Note: occasional arm breaks (arrow) were observed in ΔBLM cells. Quantification of centromere dislocation is shown below ($n = 3$ independent experiments analysing a total of 60, 60, 51 and 51 spreads in HAP1, ΔBLM, WT and Q672R cell lines; means of each experiment are shown). **e** Centromeric RPA foci formation in ΔBLM HAP1 cells expressing wild-type GFP-BLM (WT) and a BLM helicase-dead mutant (GFP-Q672R) following BI2536 treatment. Representative images showing the lack of RPA foci at centromeres in the GFP-Q672R cells. Right: bar graphs showing the average fluorescence intensities of centromere GFP and RPA foci, respectively, in GFP-BLM and GFP-Q672R cells (mean + S.E.M. is shown). A scatter plot of RPA foci intensity by GFP foci intensity at centromeres (total numbers of centromere foci analysed: GFP-BLM, $n = 582$; and GFP-Q672, $n = 481$). All RPE1 and HAP1 cells, including their derivatives, were pre-synchronised at G1/S by single thymidine block. Drugs were added at 6 h post-release. After 2 h treatment, cells were subject to immunofluorescence staining. RNAi treatment of RPE1 cells was performed for 23 h before G1/S release. Scale bars = 5 μm

activity is required. We generated polyclonal cell lines from the ΔBLM HAP1 cells, which stably express either a GFP-tagged WT or a helicase-dead (Q672R) BLM protein. The expression of the GFP-Q672R protein was similar to the endogenous BLM level in HAP1 cells; whereas, the GFP-WT was over-expressed (Supplementary Fig. 13b). In agreement with our notion, the helicase-dead (Q672R) BLM failed to induce centromere DNA threads and dislocations caused by BI2536 treatments (Fig. 6c, d). However, as the expression level of the GFP-Q672R mutant was lower than the WT control; to perform a better comparison, we re-sorted the WT GFP-BLM cells to obtain a cell population with a lower BLM expression (Supplementary Fig. 13c). Despite a much lower abundance, the WT GFP-BLM protein was still capable of driving centromere DNA thread formation (Supplementary Fig. 13d). More importantly, the GFP-Q672R mutant protein no longer induced RPA foci formation, despite its aberrant enrichment at centromeres following PLK1 inhibition (Fig. 6e). Therefore, we conclude that centromere disintegration is mediated by BLM-dependent DNA unwinding at centromeres.

Next, we investigated the role of PICH. Knockdown of PICH, like BLM, also suppressed BI2536-induced centromere DNA thread formation and centromere dislocations (Fig. 7a–c). However, it also abolished BLM localisation and RPA formation at K-chromatin. (Fig. 7d). Therefore, PICH acts upstream to facilitate the recruitment of BLM to centromeres after PLK1 inactivation. Taken together, our data suggest that PLK1 has an important function to protect centromeres from unlawful DNA unwinding, mediated by the PICH/BLM complex. The structural change probably impairs centromere rigidity and causes the failure to withstand bipolar spindle pulling forces. Consequently, centromeres are torn apart, leading to whole-chromosome arm splitting and chromosome biorientation failure.

**Centromeric tethering of BLM does not induce metaphase collapse.** Both PICH and BLM interact with PLK1 and are hyperphosphorylated during mitosis[10,39–41]. Hyperphosphorylation of PICH and BLM is partially dependent on PLK1[42] (Supplementary Fig. 14). It has been proposed that hyperphosphorylation of BLM can prevent its association with mitotic chromosomes[41,43]. Thus, we sought to test whether the abnormal loading of BLM to centromeres, presumably due to the loss of PLK1-mediated phosphorylation, might cause centromeric DNA unwinding and dislocation. We tethered BLM to centromeres in HeLa cells by fusing a truncated CENPB (1–158) to a GFP-tagged BLM. Transient expression of the WT GFP-BLM and the CENPB-GFP-BLM fusion proteins showed that the WT GFP-BLM exhibited diffused localisation pattern and was mostly excluded from mitotic chromosomes after nuclear envelope

breakdown. In contrast, the CENPB-GFP-BLM fusion protein was enriched at core centromeres throughout mitosis (Supplementary Fig. 15a). However, we did not find that tethering BLM to centromeres induced obvious mitotic defects such as mitotic arrest, as observed by PLK1 inhibition. Time-lapse live-cell imaging showed that the CENPB-GFP-BLM transfected cells, like the WT GFP-BLM, progressed successfully into anaphase, without metaphase collapse (Supplementary Fig. 15b). Moreover, RPA (ssDNA) formation was not detected at centromeres where the CENPB-GFP-BLM protein was enriched (Supplementary Fig. 15c). Therefore, artificially over-loading BLM at centromeres seems not sufficient to trigger DNA unwinding and centromere disintegration when PLK1 remains active. Though speculative, the triggering of centromere dislocation in PLK1-inactivated cells might be caused by mis-regulation of BLM (and PICH) activity; and/or because of improper formation of centromere structures that mis-activates the PICH/BLM complex prior to chromosome disjunction.

**Constitutive PLK1 activity for centromere integrity maintenance.** To test if centromere disintegration might be caused by centromere malformation during early mitosis, we inhibited PLK1 only after mitotic cells had fully formed their chromosomes and progressed into metaphase, whilst in the presence of active PLK1. RPE1 cells stably expressing a GFP-tagged PLK1 were first blocked at metaphase using the APC/C inhibitor, ProTAME. Time-lapse live-cell imaging recorded that upon the addition of BI2536, the fully bioriented chromosomes started losing their alignment. Most importantly, this was accompanied by the formation of DNA threads (Fig. 8a and Supplementary Movies 2 and 3), indicating the occurrence of centromere dislocation. Furthermore, we found that centromere dislocation can happen rapidly, as within 30 min of BI2536 addition, more than 60% of the metaphase-arrested cells generated centromere DNA threads (Fig. 8b). As centromere dislocation depends on bipolar spindle pulling, this would imply that the KT-MT attachment is not instantly destroyed, at least in those centromeres with DNA threads. Therefore, rather than due to an initial malformation, centromere disintegration is likely triggered because of a defect in centromere structure maintenance.

**Depletion of PICH and BLM prolongs metaphase alignment.** Thus far, our data indicates that apart from the proposed model of KT-MT destabilisation, a failure in centromere integrity maintenance is another cause of chromosome misalignment. To further test this, we examined if suppression of centromere dislocation, by PICH and BLM depletion, might rescue the

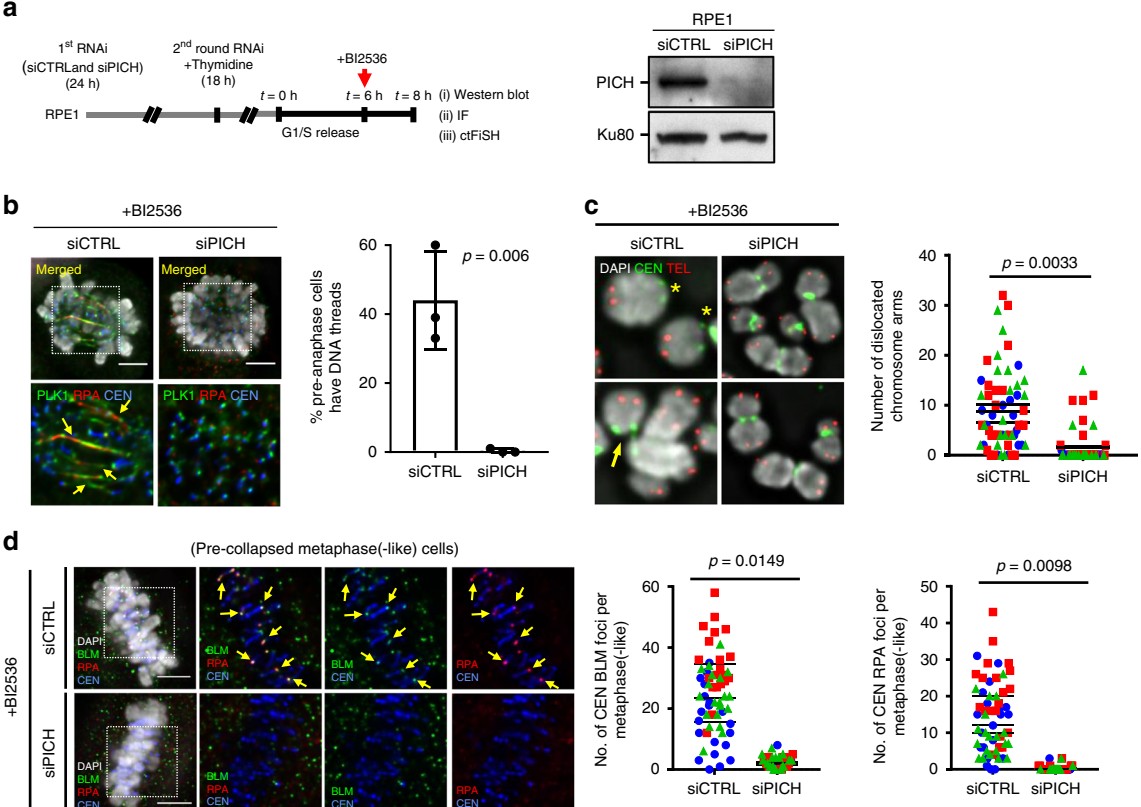

**Fig. 7** PICH acts upstream of BLM in centromere disintegration. **a** Experimental outline of RNAi depletion, in combination with thymidine synchronisation and drug treatment. Right: western blot showing PICH protein level in siCTRL and siPICH treatment. Ku80 is used as a loading control. **b** Centromeric DNA thread formation in siCTRL and siPICH cells under BI2536 treatment. Quantification of DNA thread formation in RPE1 cells after siCTRL and siPICH treatment ($n = 3$ independent experiments analysing 315 and 314 cells in siCTRL and siPICH conditions, respectively; mean ± S.D. is shown). **c** Representative images of mitotic spread chromosomes from cells prepared in (**a**) showing centromere dislocations (asterisks) and partial centromere splitting (arrow). Quantification of centromere dislocation is shown at right ($n = 3$ independent experiments analysing 64 and 61 dislocated chromosome arms in siCTRL and siPICH conditions, respectively; means of each experiment are shown). **d** Reductions of BLM and RPA foci formation at centromeres in siPICH cells after BI2536 treatment. Representative images (left) and quantifications (right) are shown ($n = 3$ independent experiments analysing a total of 60 cells in each treatment; means of each experiment are shown, Scale bars = 5 μm)

metaphase alignment defect in PLK1-inhibited cells. Knocking down PICH or BLM had no adverse effect on metaphase establishment in RPE1 cells under BI2536 treatments. However, it significantly prolonged the metaphase(-like) stage as compared to control cells (Fig. 8c and Supplementary Movies 4–6). Although the metaphase chromosomes inevitably misaligned after long delays in the PICH/BLM-depleted cells, they dispersed more like a 'polo' pattern, rather than the 'Fig-8' collapsed shape. As we showed that PICH/BLM depletion abolished centromere dislocations, we believe that the ultimate alignment failure is likely caused by KT-MT destabilisation. Nevertheless, it seems that even in the absence of PLK1 activity, the centromeres and KTs remain competent to support chromosome biorientation, at least in RPE1 cells, as long as the PICH/BLM complex is inactivated. Moreover, our data also implies that KT-MT destabilisation, if it occurs, seems to do so at a relatively slow rate as compared to centromere disintegration.

In summary, we report an unexpected role of PLK1 during chromosome biorientation, which prevents centromeres from destruction, mediated by the co-action of DNA unwinding by BLM helicase and bipolar spindle pulling (Fig. 9).

## Discussion
One of the key mitotic functions of PLK1 is to promote stable attachments between spindle MTs and KTs[6]. In the current study,

we reveal a hitherto undescribed role of PLK1 as a centromere guardian for chromosome alignment. We show that the lack of PLK1 activity leads to the failure of centromeres to withstand bipolar spindle pulling tension. As a consequence, centromere chromatin is stretched into a threadlike structure, resulting in centromere splitting, whole-chromosome arm separation and loss of metaphase alignment. Further experiments demonstrate that the disintegration of centromeres is not a passive process, but is actively driven through illegitimately unwinding of centromeric DNA by the PICH/BLM complex. Our results highlight a PLK1-dependent pathway for centromere maintenance during mitosis.

BLM is the key molecular driver of centromere disintegration, but it remains unclear how PLK1 counteracts its mediated destruction. Given that both BLM and PICH proteins are substrates of PLK1, a reasonable speculation is that PLK1 can regulate the activity of PICH/BLM complexes during mitosis. Previous studies have shown that before anaphase onset, BLM poorly associates with mitotic chromosomes[41,43] and on UFBs generated from prematurely disjoined sister chromatids[11]. The chromatin exclusion of BLM, presumably by hyperphosphorylation, could limit its DNA transaction activity. However, artificially tethering BLM to centromeres is not sufficient to induce DNA unwinding and centromere dislocation, which may suggest that either the BLM protein remains inactive, or additional factors such as PICH activation and/or chromatin

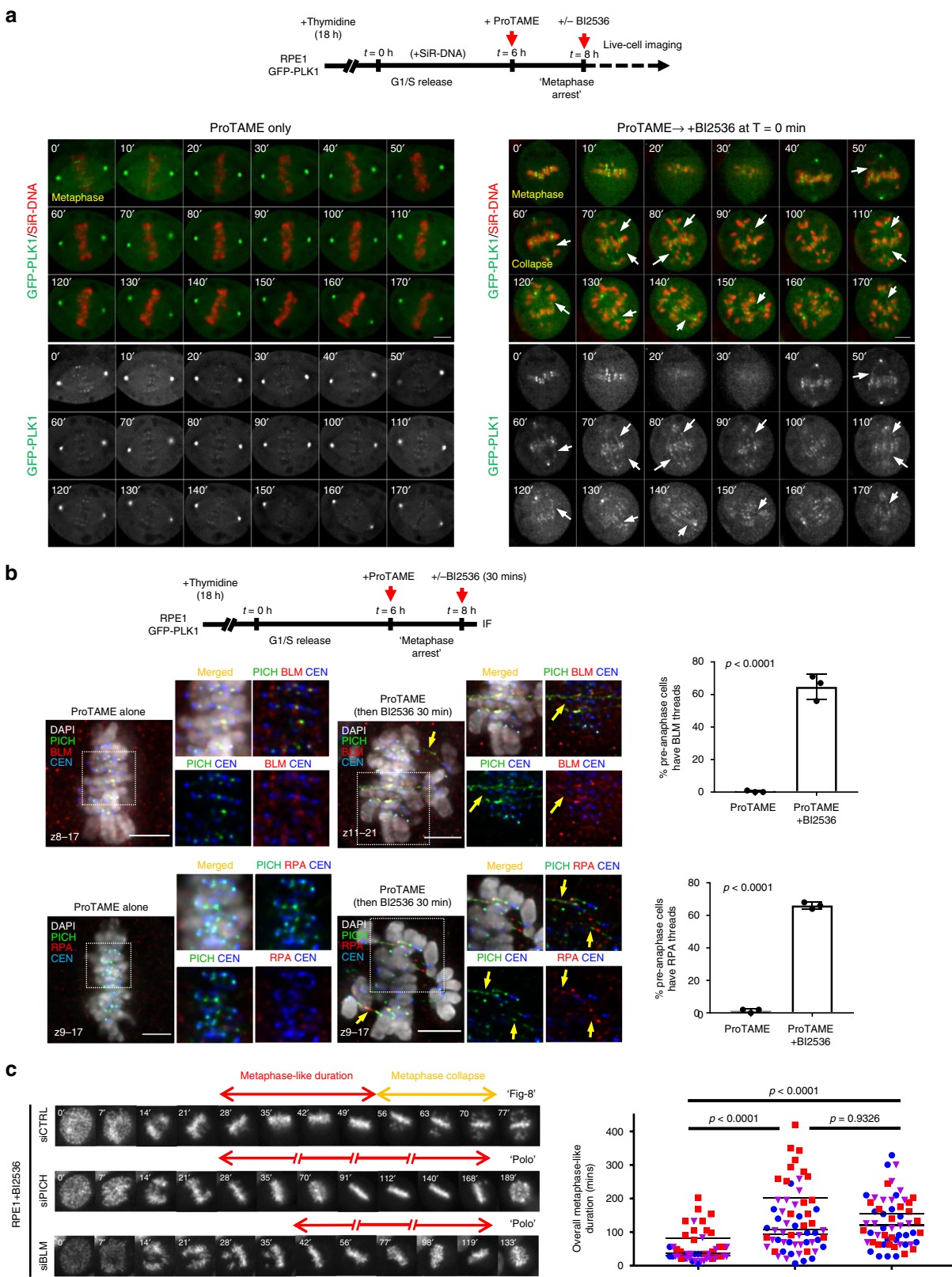

remodelling are required. Alternatively, PLK1 may protect centromeres through facilitating normal condensation of centromeres, a process if compromised might create a DNA substrate that mis-activates the PICH/BLM complex. However, impairing chromosome condensation by condensin depletion, which leads to abnormal stretching of sister centromeres[44,45], does not trigger

similar phenotypes of centromere rupture and chromosome misalignment as induced by PLK1 inhibition. In addition, the fact that centromere disintegration can be induced in 'mature' mitotic cells; namely those cells that have fully formed normal metaphase, would suggest that rather than due to an initial chromatin malformation, it is probably caused by centromere maintenance

**Fig. 8** Constitutive PLK1 activity suppresses centromere disintegration. **a** Experimental outline and time-lapse live-cell images of GFP-tagged PLK1 RPE1 cells treated with BI2536 after metaphase establishment. Cells were arrested at metaphase by ProTAME after G1/S release. High-resolution movies were recorded immediately after the addition of BI2536. The formation of DNA threads is revealed by GFP-PLK1 protein (arrows). **b** Experimental outline and quantification of DNA thread formation in metaphase-arrested RPE1 cells. BI2536 was added in ProTAME-arrested metaphase cells for 30 min. Centromeric DNA threads were labelled by PICH, BLM and RPA staining. BLM thread counting ($n = 3$ independent experiments analysing a total of 188 and 189 cells in ProTAME and ProTAME + BI2536 conditions, respectively. RPA thread counting ($n = 3$ independent experiments of a total of 174 and 187 cells in each condition; mean ± S.D. is shown). **c** Depletion of PICH or BLM, prolongs the metaphase-(like) stage of RPE1 cells under PLK1 inactivation. Time-lapse microscopy images showing the mitotic progression of RPE1 cells treated with the indicated siRNA oligos, and BI2536. Red bars indicate the metaphase(-like) stage; yellow bars indicate 'metaphase collapse'. Quantification (right) of the overall duration of metaphase(-like) stage in control, PICH- and BLM-depleted cells, following BI2536 treatment ($n = 3$ independent experiments analysing 50, 60 and 57 cells in siCTRL, siPICH and siBLM conditions; means of each experiment are shown). Scale bars = 5 μm

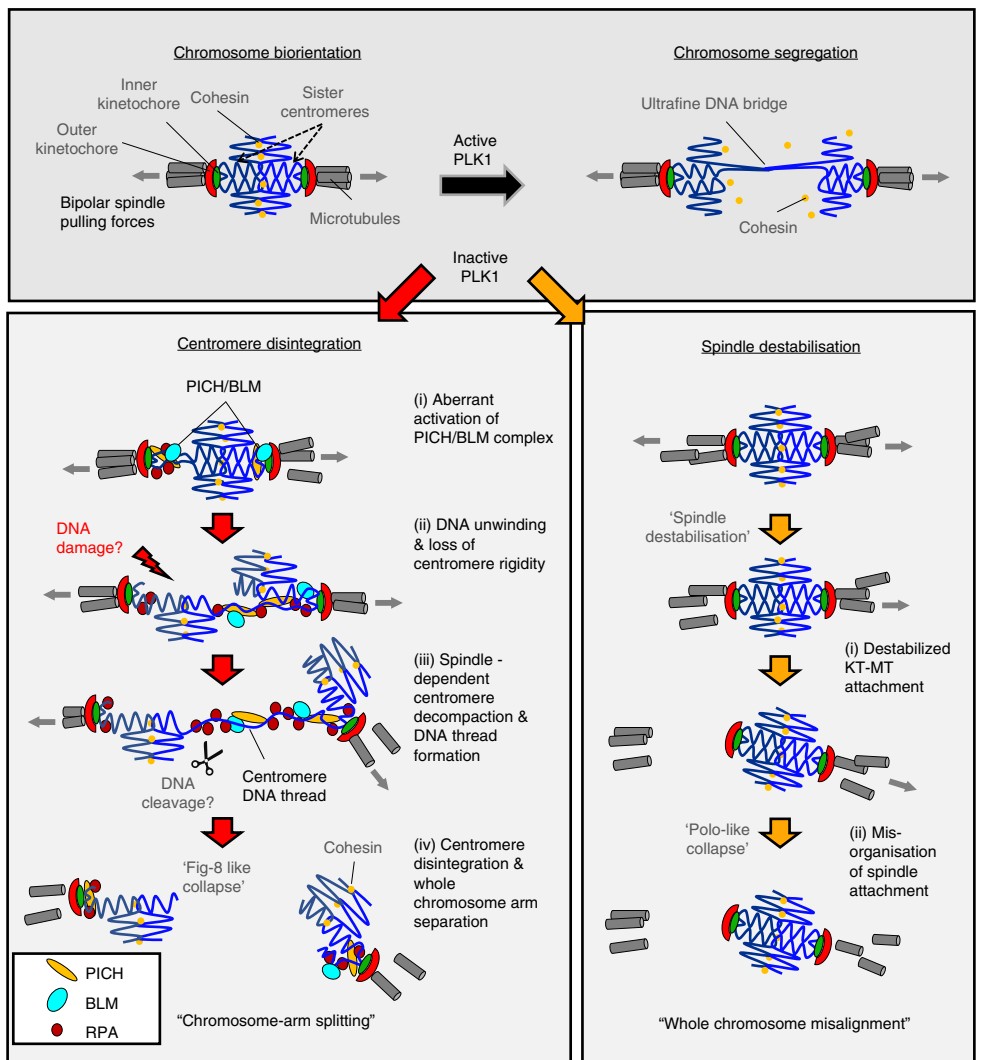

**Fig. 9** A model of centromere disintegration in chromosome misalignment. Active PLK1 is required to stabilise chromosome biorientation for chromosome segregation. In addition to the existing role for spindle stabilisation, our current study demonstrates that PLK1 also functions to maintain centromere integrity for chromosome alignment. In the absence of PLK1 activity, centromeres are aberrantly targeted by BLM helicase in a PICH-dependent manner. It leads to unlawful, excessive formation of ssDNA, which impairs centromere configuration and weakens it ability to withhold kinetochore complexes. Forces exerted by the bipolar spindle attachment pull out the centromere chromatin, which might trigger further DNA unwinding by the PICH/BLM complex. As a consequence, it decompacts the centromere axis, leading to the formation of centromere DNA threads and whole-chromosome arm separation. Cells therefore fail to maintain chromosome biorientation and result in metaphase collapse with a 'Fig-8' like misalignment pattern. Alternatively, if the spindle attachment is destabilised first, it prevents centromere dislocation but this will lead to whole-chromosome misalignment, which probably manifests as more like a 'Polo' pattern. Therefore, we propose that PLK1 plays a multifactorial role in establishing and maintaining chromosome biorientation by both protection of centromere integration and stabilisation of KT-MT attachment

impairment. Nevertheless, no matter if there is a structural defect, loss of PLK1 does not seem to greatly compromise chromosome biorientation, at least in RPE1 cells, as long as the PICH/BLM complex is inactivated. We postulate that PLK1 may protect centromeres through both chromatin structure maintenance and the regulation of the PICH/BLM complex activity. Further experiments will need to dissect the underlying mechanism(s).

Another very intriguing finding is that when acting in concert with bipolar spindle tension, the PICH/BLM complex can promote decompaction of the centromere axis. This converts the centromeric chromatin into an ultrafine DNA structure, reminiscent of anaphase UFBs[10,11], leading to whole-chromosome arm separation. Conceivably, if such decompaction activity is applied at a chromosomal region where sister DNA intertwinements persist, it may be able to relieve the entangling constraints and facilitate the poleward separation of sister chromatids during anaphase. Though this is speculative, this finding could provide an alternative clue to understand how the UFB-binding complex may function during chromosome disjunction, and potentially also explain why a long region of UFBs is always coated by the PICH/BLM complex. In principle, such powerful action would need to be under a tight control before anaphase onset, otherwise it could lead to pathological damage at chromatin sites where tension is exerted; namely the centromere. Finally, the identification of a centromere-specific breakage pathway, independent of chromosome mis-segregation[46,47], also offers an alternative direction in understanding the origin of complex chromosome rearrangements, such as whole-chromosome arm rearrangements, which are observed in many human tumours and rare genetic disorders[48–51].

In conclusion, our study unveils an unexpected participation of PLK1 and the UFB-binding complex in the safeguard of centromere integrity during mitosis, which is critical for faithful chromosome segregation and chromosome stability.

## Methods

**Cell culture**. RPE1-hTERT, 82-6-hTERT normal diploid cell lines, 1BR3 primary fibroblasts, HCT116 colon and HeLa cancer cells were obtained from the Genome Damage and Stability Centre (GDSC) Cell Bank. All cell lines were authenticated by STR genotyping from European Collection of Cell Cultures. RPE1-hTERT derivative cells were generated and supplied by Mark Burkard (University of Wisconsin). Bloom's syndrome fibroblasts (GM08505) were obtained from Phillip North (University of Oxford). HAP1 cells and HAP1 ΔBLM cells were obtained from Marcel van Vugt (University of Groningen). All cell lines passed mycoplasma tests (Lonza MycoAlert kit). RPE1-hTERT and its derivative cells were grown in DMEM/F-12 medium (Sigma) containing 15% foetal calf serum (FCS) and Pen/Strep antibiotics (P/S). 82-6 Fibroblast cells were grown in DMEM/F-12 medium containing 15% FCS and P/S. HAP1 cells were grown in IMDM (Gibco) containing 10% FCS and P/S. 1BR3 primary cells were grown in MEM (Gibco) containing 2mM L-glutamine, 15% FCS and P/S. HCT116 cells were grown in McCoy's 5A (Gibco) containing 15% FCS and P/S. Bloom's syndrome fibroblasts (GM08505) were transfected with a pEGFP-hBLM construct and selected by 700µg/ml G418 for 14 days. A single clone was isolated and maintained in MEM (Gibco) containing 2 mM L-glutamine, 10% FCS, P/S and G418. Cell cultures were maintained at 37 °C in a humidified atmosphere containing 5% $CO_2$. GFP-BLM(WT) and GFP-BLM(Q672R) HAP1 cells were generated by stable transfection with the corresponding constructs in HAP1 ΔBLM cells by using FuGene HD (Promega) according to the manufacturer's guidelines. The DNA constructs were created by sub-cloning EGFP-hBLM (WT) or Q672R (helicase dead mutant) fragments into a pSYC-181-(NEO) vector. Following a 1.2 mg/ml of G418 selection for 14 days, GFP-positive populations were sorted and isolated using a FACS cell sorter (BD FACSMelody).

**Cell synchronisation and drug treatments for mitotic cell analysis**. Cells were treated with 2 mM of thymidine for 18 h to enrich cells at the G1/S boundary. Cells were then released into S-phase by washing three times with pre-warmed culturing medium, or pre-warmed 1× PBS and released into fresh medium. Totally, 5–6 h post-G1/S release, indicated inhibitors were added. At approximately 8–9 h post the G1/S release, mitotic cells were fixed or enriched for analyses.

**RNA interference**. Cells were transfected with siRNA oligonucleotides using Lipofectamine RNAiMAX transfection reagent (Thermo Fisher Scientific) following the manufacturer's guidelines. Cells underwent 1 or 2 rounds of siRNA transfection as necessary.

Non-targeting siRNA pool (Dharmacon ON-TARGET plus Non-targeting Pool—D-001810-10-05. UGGUUUACAUGUCGACUAA; UGGUUUACAUGUUG UGUGA; UGGUUUACAUGUUUUCUGA; UGGUUUACAUGUUUUCCUA)

PLK1 siRNA sequence (Dharmacon ON-TARGET plus SMARTpool—L-003290-00-0005. GCACAUACCGCCUGAGUCU; CCACCAAGGUUUUC GAUUG; GCUCUUCAAUGACUCAACA; UCUCAAGGCCUCCUAAUAG)

Sgo1 siRNA sequence (Dharmacon ON-TARGET plus SMARTpool—L-015475-00-0005. CAGCCAGCGUGAACUAUAAA; GUUACUAUCUCACAU GUCA; AAACGCAGGUCUUUUAUAG; GUGAAGGAUUUACCGCAAA)

BLM siRNA sequence (Dharmacon ON-TARGET plus Individual—J-007287-08-0005. GGAUGACUCAGAAUGGUUA)

PICH siRNA sequence (Invitrogen—AAUUCGGUAAACUCUAUCCAC AGCU)

**Fluorescence immunostaining**. For immunostaining analyses, cells were seeded onto No. 1.5 or No. 1.5H cover glass and fixed with Triton X-100-PFA buffer (250 mM HEPES, 1× PBS, pH7.4, 0.1% Triton X-100, 4% methanol-free paraformaldehyde) at 4 °C for 20 min, or with PBS–PFA buffer (1× PBS, 4% methanol-free paraformaldehyde) at room temperature for 10 min. Pre-extraction was carried out in indicated experiments before fixation by incubation of the cover glass in pre-extraction buffer (20 mM HEPES pH7.4, 0.5% Triton X-100, 50 mM NaCl, 3 mM $MgCl_2$, 300 mM sucrose) for 10–15 s. Cells were incubated in permeabilisation buffer (0.5% Triton X-100, 1XPBS) for 20 min on ice followed by blocking with foetal calf serum for 15 mins at room temperature. Cells were incubated with primary antibody at 37 °C for 90 min followed by secondary antibody incubation at room temperature for 30 min. Slides were washed with 1× PBS for 5 times at room temperature after antibody incubation. Cells were mounted using DAPI-containing Vectashield mounting medium.

Primary antibodies used: anti-PICH (Abnova; H00054821-B01P, 1:100), anti-PICH (Abnova; H00054821-D01P, 1:100), anti-BLM (Santa Cruz; sc-7790, 1:50), anti-BLM (Abcam; ab2179, 1:200), anti-γH2AX (Upstate; JBW-301, 1:400), anti-TOP2A (Santa Cruz; sc-5348, 1:100), anti-SMC2 (Bethyl Lab; A300-058A, 1:200), anti-RPA70 (Abcam; ab79398, 1:200), anti-RPA32 (Abcam; ab2175, 1:200), anti-CENPA (Abcam; ab13939, 1:100), anti-CENPB (Abcam; ab25734, 1:800), anti-NUF2 (Abcam; ab122962, 1:200), anti-PLK1 (Santa Cruz; sc-55504, 1:100), anti-pericentrin (Abcam; ab4448, 1:400), anti-centromere (ImmunoVision; HCT-0100, 1:400) and GFP booster (ChromoTek; gba-488, 1:200). Secondary antibodies used: donkey anti-mouse Alexa Fluor 488, 555 and 647; donkey anti-rabbit Alexa Fluor 488, 555 and 647; donkey anti-goat Alexa Fluor 488 and 555; goat anti-human DyLight 550 and 650 (All secondary antibodies are purchased from ThermoFisher and used at 1:500 dilution).

**High-resolution deconvolution microscopy**. Images were acquired under a Zeiss AxioObserver Z1 epifluorescence microscopy system with 40×/1.3 oil Plan-Apochromat, 63×/1.4 oil Plan-Aprochromat and 100×/1.4 oil Plan-Aprochromat objectives and a Hamamatsu ORCA-Flash4.0 LT Plus camera. The system is calibrated and aligned by using 200 nm-diameter TetraSpeck microspheres (T7280, ThermoFisher). Ten to fifty z-stacking images were acquired at 200 nm intervals covering a range from 2 to 10 µm by using ZEN Blue software.

Deconvolution was carried out using Huygens Professional deconvolution software (SVI) with a measured point-spread-function generated by 200 nm diameter TetraSpeck microspheres. Classical maximum likelihood estimation method with iterations of 40–60 and signal-to-noise of 20–60 was applied.

**Time-lapse Live-cell microscopy**. Cells were seeded on 2-well or 4-well tissue culture chambers coverglass II (Sarstedt). SiR-DNA (Spirochrome) was added for at least 5 h prior to live-cell imaging. Images were acquired under a Zeiss AxioObserver Z1 epifluorescence microscopy system equipped with a heating and $CO_2$ chamber (Digital Pixel) by using 40×/0.6 Plan-Neofluar or 40×/1.3 oil Plan-Apochromat objectives and a Hamamatsu ORCA-Flash4.0 LT Plus camera. For mitotic progression analysis, 5–10 z-stacking images with 2 µm intervals were taken with the indicated time intervals by using ZEN Blue software. Images were processed using ImageJ software and in-focus z-plane images were manually extracted to make image montages. For imaging of DNA thread formation in live cells, 40×/1.3 oil Plan-Apochromat objective was used to capture eight z-stack images with 800 nm intervals and in-focus z-plane images were extracted using ImageJ software.

**Chromosome spread preparation**. Following synchronisation using thymidine, cells were treated with pre-warmed hypertonic solution for 5–10 min at 37 °C (0.075M KCL). The swollen cells were then fixed and washed twice with methanol: acetic (3:1 ratio), before finally being re-suspended in fresh methanol:acetic solution. Chromosome spreads were dropped onto glass slides and either counterstained with Vectashield plus DAPI, or stored at room temperature for

forthcoming FISH hybridisation. Colcemid was omitted in all mitotic spread preparations.

**Centromere and telomere peptide nucleic acid (PNA) FISH**. Centromere (CENPB-FAM; PNABio) & Telomere (Tel-Cy3 PNA FISH kit; DAKO, Agilent) PNA probes were hybridised according to the manufacturer's instructions. Briefly, chromosome spreads were rehydrated in 1× TBS prior to fixation in 3.7% PFA solution. Slides were then washed and pre-treated before dehydration using a gradient ice-cold ethanol wash (70, 90 and 100%). Slides were air dried and PNA probes were added. Slides were then co-denatured at 80 °C for 1 min and incubated for 2 h at room temperature. Slides were then washed in FISH Wash solution (Tel-Cy3 PNA FISH kit; DAKO, Agilent) for 5 min at 65 °C following by dehydration using a series of ethanol wash before counterstaining using DAPI Vectashield.

**Multi-colour FISH**. mFISH was performed by using 24XCyte Human Multicolour FISH probe (MetaSystems) according to the manufacturer's instructions. Images were acquired by MetaSystems using a Zeiss AxioObserver Z1 epifluorescence microscopy system with a CoolCube CCD camera and 100×/1.4 oil Plan-Aprochromat objective. Multi-colour FISH (mFISH) karyotyping was carried out by using ISIS Imaging software.

**Immunoblotting**. Cells were trypsinized and lysed on ice for 15–20 min with lysis buffer (50 mM Tris pH 7.5, 300 mM NaCl, 5 mM EDTA, 1% Triton X-100, 1.25 mM DTT, 1 mM PMSF and cOmplete™ protease inhibitor cocktail). Protein concentration was quantified using a Bradford assay (Bio-Rad). Immunoblotting (IB) was performed following standard procedures. Primary antibodies used for IB in this study: anti-BLM (Abcam, ab2179, 1:2000), anti-PICH (Abnova; H00054821-B01P, 1:300), anti-GFP (Abcam, ab290, 1:1000) and anti-Ku80 (Abcam, ab80592, 1:10000). All uncropped blot scans are available in the Supplementary excel data file.

**Flow cytometry**. Cells were trypsinised, washed with PBS and fixed with 70% ice-cold ethanol. For cell cycle analysis, cells were washed with PBS and re-suspended in propidium iodide/RNaseA staining buffer. FACS profiles were then determined and analysed using BD Accuri C6 sampler.

**KT/centromere foci measurement**. Samples were subjected to pre-extraction in pre-extraction buffer (20 mM HEPES pH7.4, 0.5% Triton X-100, 50 mM NaCl, 3 mM MgCl$_2$, 300 mM sucrose) for 10–15 s followed by fixation and immuno-fluorescent staining as described above. Thirty to fifty z-stacking images with 200 nm intervals were acquired and deconvolved using Huygens Professional decon-volution software (SVI). KT foci on each single z-plane were marked and measured using the ImageJ Plugins detailed below.

**ImageJ measurement of KT foci coordinates, distances and intensities**. Spot Pair Distance Tool: Measures the distance between spots in two channels of an image. The tool searches within a focus/box radius, typically± 5px, for a local maxima in the two pre-selected analysis channels. The centre-of-mass around each maxima, typically± 2px, is computed as the centre of intensity for each channel. Dragging from the clicked point creates a reference direction. The Euclidean distance between the centres is reported, optionally with the signed XY distance and angle relative to the reference direction. Visual guides are overlaid on the image to assist in spot selection and direction orientation. Available in the latest GDSC ImageJ plugins.

Spot Fit Tool: Fits a 2D Gaussian to a spot in an image. The tool searches within a box radius, typically± 3px, for a local maxima in the pre-selected analysis channel. A 3 × 3 smoothing filter is applied before identification of the maxima. A 2D Gaussian function is then fitted to the data using non-linear least-squares fitting and poor fits rejected using a signal-to-noise ratio. The parameters for the fit are reported including the total intensity under the Gaussian function and the local background value. Visual guides are overlaid on the image to show the fitted location. Available in the pre-release GDSC SMLM ImageJ plugins.

**Statistics**. Statistical analysis was performed using GraphPad Prism 7 software by two-tailed unpaired Student's $t$ test and two-way ANOVA as per the experimental requirement.

**Recombinant DNA and transfections**. CENPB (1–158aa) cDNA fragment was PCR amplified from a PLK1 plasmid in which the C-terminal PBD domain was replaced with the first 158 amino acids of CENPB (pQCXIN-Flag-Plk1deltaC-CENPB(1–158)) (a gift from Mark Burkard) and cloned into full length pEGFP-hBLM and pEGFP-hBLM(Q672) plasmids at AgeI site to generate N-terminally tagged CENPB (1–158aa) fusion proteins. Transfections of DNA plasmids were performed using FuGene HD (Promega) according to the manufacturer's guide-lines. All plasmids and their sequences are available upon request. Forward primer: (CENPB-For1) 5′-TAAGCAACCGGTATGGGGCCCCAAGAGGCGACAG-3′;

Reverse primer: (CENPB-linker-Rev1)5′-TAAGCAACCGGTCTAGCACTT GCGCCCCCAGCACTTGCTCCACCGGCCGGACTG GCAGGCGCCCGC-3′

**Reporting summary**. Further information on research design is available in the Nature Research Reporting Summary linked to this article.

## Data availability

The authors declare that the data supporting the findings of this study are available within the paper and its Supplementary information files. Raw imaging data are available from the corresponding author upon reasonable request.

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

## Acknowledgements

We would like to thank the people in the Genome Centre for their great support. We thank Robert Lera and Mark Burkard for providing the RPE1 derivative cells and the CENPB cDNA, Marcel van Vugt for HAP1 and HAP1 BLM-knockout cells, and Phillip North for GM08505 Bloom Syndrome fibroblasts. We thank Jessica Hudson for BLM siRNA oligos. We also thank Kim Nasmyth, Jon Baxter and Mark Burkard for helpful comments on the study. This work is supported by Sir Henry Dale Fellowship (Ref. 104178/Z/14/Z) from Wellcome Trust and the Royal Society, and by the Genome Damage and Stability Centre. K.L.C. is the recipient of Sir Henry Dale Fellowship. Funding for open access charge: Charity Open Access Fund (COAF).

## Author contributions

O.A.J. and K.L.C designed and performed the experiments with help from A.T., T.O. and A.H. O.A.J, A.T., T.O. and K.L.C. analysed the data. K.L.C. wrote the paper with inputs from all authors.

## Additional information

**Competing interests:** The authors declare no competing interests.

