## [Peer Review File · Nature Communications]

Reviewers' comments:

Reviewer #1 (Remarks to the Author):

The Plk1 cell cycle kinase has a number of important roles during cell division including the control of mitotic entry and bipolar spindle formation, the promotion of proper microtubule-kinetochore attachments and the orchestration of cytokinesis. Loss of Plk1 activity or the PLK1 molecule altogether results in spindle checkpoint mediated cell cycle arrest with monopolar spindle. Jones et al., investigate the role of PLK1 in promoting metaphase by inhibiting Plk1 activity using the well-characterised small molecule inhibitor BI2536. In contrast to previous reports they find that an aligned metaphase state is actually transiently achieved – at least in untransformed cells – but is then lost. Careful investigation of the Plk1-inhibitor treated cells revealed a loss of centromere integrity with a concomitant occurrence of PLK1-, PICH- and BLM-coated ultrafine DNA threads in the centromeric region (referred to as pre-anaphase inter-chromatin DNA threads (PITs)). In further experiments the authors report that this destabilization of the centromeric DNA leads to a splitting of chromosomes at the centromere into p- and q-arms. The authors refer to this event as “centromere dislocation”. Both the formation of the abnormal chromosome threads and the centromere splitting are dependent on the presence of PICH and BLM. BLM appears to be the key enzymatic activity for this event, whereas PICH seems to be mainly required to localize BLM. Altogether, the authors suggest that one crucial function for Plk1 during mitosis is to prevent untimely unravelling of the centromeric DNA due to uncontrolled BLM activity.

This study reports a very interesting novel aspect of PLK1 function during mammalian mitosis which will be of interest to a wide readership. The idea that PLK1 activity is required during mitosis to prevent unravelling of the centromere is particularly interesting. The experiments are conducted to a very high standard and the results are generally convincing and well presented. The major shortcoming of this manuscript is that it is mainly descriptive and does not give any insight into how Plk1 may be exerting this novel, centromere-protective function. I don't feel that this should preclude publication of these interesting observations, but the manuscript would certainly be improved by checking the obvious possibility that BLM is a target of Plk1. Is BLM phosphorylated by Plk1? If so, could this modulate BLM activity? Minimally, an experiment addressing whether BLM is a Plk1 target or not (this could be done by Western blotting of BLM in extracts treated with vehicle control or BI2536) would nicely round off the manuscript.

The other slightly puzzling issue is that BI2536 has been used on cells for over a decade now, and the phenotype that Jones et al. describe has so far apparently been overlooked. Is this because most of the work on Plk1 has been conducted in HeLa cells? Do HeLa cells not develop the dramatic centromere splitting that Jones et al. see in RPE1 cells or are other defects, such as the bipolar spindle collapse, predominant in HeLa cells and mask the centromere splitting? It would be helpful to have a side-by-side comparison for some of the key experiments for HeLa and RPE1 cells (e.g. the live cell imaging).

Lastly, I find the terminology of “centromere delocalization” not very accurate. As far as I can see the centromere is split and not delocalized, so the correct description should be “centromere splitting” or “chromosome splitting”. Otherwise it's very confusing for the reader. The same goes for “PIT”. With that name one automatically thinks of clathrin-coated pits and membrane trafficking. When reading the manuscript I had to go back several times to remind myself what was meant by this term. I am not sure why these structures have to be referred to by an acronym – “abnormal DNA bridges” seems perfectly adequate.

Reviewer #2 (Remarks to the Author):

Chan and co-workers report an analysis of the effects of perturbing PLK1 on centromere integrity. They reveal new phenotypes arising following PLK1 inactivation with chemical inhibitors. Some of these effects are apparently associated with an aberrant activity of the Bloom helicase. Overall, the

work is conducted to a high standard, the subject matter is of significant interest to cell biologists, and the data are generally convincing. However, there are some weaknesses and the study as it stands fails to establish causative connections between key events.

Specific remarks:

1. I found the tone of the text far too dogmatic (things are stated as being 'proved' etc) and in places seriously overstated. This begins with the title, but there are also a lot of examples in the body of the manuscript. In particular, the claim in the title that BLM mediates dechromatinization is misleading in my view and should be dropped.
2. The summary states that BLM is 'illegitimately recruited'. My understanding from the literature is that BLM is found at centromeres anyway. What is the evidence in the current study for it being an illegitimate event?
3. A fundamental issue with studies of this sort where a kinase that plays multiple roles in a process is inhibited, is that it is very hard to dissect whether the apparent phenotypes are primary, secondary or tertiary effects of loss of the function of the enzyme. The structure of the kinetochore or centromere might be fundamentally different in the inhibitor treated cells, which then means that structures accumulate that don't exist in control cells. Hence, everything downstream of that becomes artefactual and doesn't reflect an event that would ever occur in 'real life'. Some more balanced discussion reflecting the inability to distinguish primary from secondary events would be appropriate.
4. On a related note, the authors failed to demonstrate any functional connection between PLK1 and BLM. Without that, it is hard to be sure that they are not studying phenomenology that has no relation to cell physiology. It is important to make that connection and to determine how BLM is inhibited by PLK1.
5. Is there a need to call the claimed 'new' DNA structure by a specific name? The PITs show the same characteristics as UFBs, which are also visible at centromeres before anaphase (shown by the Nigg group). What is the evidence that PITs are anything fundamentally different from UFBs?
6. The claim that spindle pulling forces cause the stretching and other distortions of the DNA is not fully demonstrated by the data. At best, nocodazole partially suppresses the effects of PLK1 inhibition. For example, in Figure 2b, statistical analysis of Noc versus BI+Noc should be performed. They look very different to me. If so, that would change the strength of any conclusion.
7. Could the authors comment on how spindle pulling could cause longitudinal stretching of the chromosome?
8. The authors rightly state that PLK1 is not needed for PICH to localize to centromeres. However, the Nigg group showed that PLK1 depended on PICH. If true that would really complicate the picture. I feel that this issue needs to be investigated.
9. RPA forms a complex with BLM and therefore using RPA to define ssDNA might be complicated by the fact that it might be recruited to centromeres through binding BLM. A second method to detect ssDNA would be valuable (native BrdU?).
10. The experiments with the Q672R mutant of BLM are compromised by the fact that the mutant protein is expressed at a significantly lower level than the control cells. This is fine if the mutant complements the defect, but it doesn't in this case. A better pair of matched clones needs to be analyzed.

Reviewer #3 (Remarks to the Author):

The Polo like kinase 1 (Plk1) is one of the most studied kinase mainly required to regulate chromosome segregation during mitosis. Inhibition of Plk1 activity is known to affect phosphorylation of several key components of the mitotic network leading to mitotic arrest and cell death.

In this manuscript, the authors described the consequences of PLK1 inhibition on metaphase chromosomes. They first observed that PLK1 inhibition in metaphase leads to the formation of a new

type of DNA structure that they called 'pre-anaphase interchromatin DNA threads' (PITs). This structure somehow resembles the previously identified Ultrafine anaphase bridges (UFBs). They proposed that these structures result from the illegitimate recruitment of BLM at the centromeres under conditions of PLK1 inhibition. Due to the mitotic spindle forces, these bridges ultimately fail to withstand spindle tension, resulting in whole chromosome arms splitting.

The authors performed a large amount of work and described several and novel outstanding phenotypes that are certainly of interest to the community and the reader of Nature Communications. Nevertheless, the interpretation of some of these observations remain poorly established and some key experiments are missing to test their model. In addition, several main points are contradictory with the literature and have to be addressed before publication in Nature Communications.

Major points:

1) BLM protein has been shown to be recruited from G2 to mitosis at the centromere leading to changes in centromeric structure together with topoisomerase IIa. Their cooperation is required to resolve centromeric UFB. BLM depletion is associated with several mitotic abnormalities, including UFB and chromatin bridge formation as described by the same corresponding author some years ago (Chan et al., 2007, Baumann et al., 2007, Chan and Hickson, 2011). So, it is well established that BLM activity is necessary to resolve abnormal DNA structures during mitosis, thus preventing genetic instability.

Here, the authors proposed that, during an unperturbed mitosis, BLM activity is inhibited by PLK1 in order to maintain centromeric integrity, therefore preventing PITs formation (a DNA structure very similar to the UFBs). Their data imply that BLM depletion rescues drastic chromosomal defects as PITs formation and loss of centromere integrity, which is very surprising based on the previously described role of BLM. The authors should explain this discrepancy with the literature. Indeed, it is difficult to imagine why BLM is recruited at the centromere during mitosis if its function leads to such catastrophic events in the absence of PLK1 activity.

2) Most of the experiments have been done upon several synchronization events rendering the interpretation of the results difficult. For example, abnormal nucleotide pool concentration is known to induce replication stress and so thymidine treatment could explain some phenotypes observed by the authors. It is not clear to this reviewer why the authors had to synchronize cells at all as they specifically look at cells during mitosis, so they could perform all their experiments in unsynchronized and unperturbed cells. The authors have to show that centromeric dechromatinization and disintegration arise in unsynchronized cells as well (and not with 16hr of CDK1 inhibitor as they did in sup figure 3c) eliminating thus a potential role of the synchronization in the phenotypes described in this paper.

More importantly, most of the phenotypes described in this paper result from PLK1 inhibition during 8 hours (cf Figure 1a), whereas mitotic duration in RPE cells is estimated at 50 minutes. Thus, in this condition the mitotic duration is extended by 8 times rendering it difficult to appreciate the biological significance of the subsequent phenotypes. The authors have to discuss this point. On this regard, it is unclear which control the authors used. Which phase is compared here in DMSO (e.g. Figure 1e, but this is true for many of all other figures)? As time in mitosis is very different between cells with PLK1 inhibition and cells treated with DMSO, it is better to have a similar control of mitotic phase, for example by inhibiting CENP-E.

3) Topoisomerase IIa has been extensively described to promote the resolution of centromeric DNA structures during mitosis (Porter and Farr, 2004), in association with PICH and BLM (Nielsen et al., 2015; Rouzeau et al., 2012). Given the importance of this protein in the processing of centromeric DNA, the authors have to show whether topo IIa activity is involved in PIT formation.

4) To test the implication of the spindle forces on PIT formation the authors used Nocodazole treatment that leads to microtubule depolymerization. This treatment can test a direct role of microtubules but cannot be used to directly analyze the implication of mitotic spindle forces. A better way to do that would be to use Eg5 inhibitor to inhibit outward forces or dynein inhibitor to inhibit inner forces. The authors should explore the role of the main drivers of mitotic spindle forces to exclude an implication of these forces in PIT formation.

- 5) The authors conclude that PITs correspond to DNA structures based on RPA staining (and presumably binding). This is indirect proof. The authors should use DNase treatment or BrdU incorporation (as previously done for UFB characterization (Chan et al., 2007)) to clearly establish that PITs are DNA structures.
- 6) BLM protein mainly acts in a complex with Topo III / RMI1 / RMI2 forming the BTRR complex (Manthei and Keck, 2013). This complex is crucial to resolve linked DNA intermediates. More importantly, recently a new model implicating the BTRR complex in the resolution of DNA structure mimicking DNA threads (comparable to UFB or PIT structures) has been proposed (Sarlós et al., 2018). It will be of interest (but not absolutely required) if the authors could test and discuss if the BTRR complex is also involved in PIT formation/resolution.
- 7) Figure 2b, chromosomes look very weird and not convincing. All chromosome regions seem to be affected by the inhibition of PLK1, not only centromeres. Is this due to loss of PLK1 activity or prolonged mitosis or simply due to technical problems with this spread+FISH. As the chromosomes look normal in other experiments done w/o FISH, the last option seems to be the most likely. The authors should try to perform an immunofluorescence coupled with chromosome spread and stain for telomeres and centromeres to confirm their findings.
- 8) To better understand if centromeres break at centromeres or at their proximity, the authors should measure CENP-A and CENP-B boxes (or CENP-B) amount. If the breaks are at centromeres, intensity of CENP-A/B will drop. If not, this would suggest that breaks do not occur at centromeres. Another approach that the authors can use is to perform chromosome spreads using specific centromeric probes to see if they observe centromeric signal at both sides of a broken chromosome. It is unclear from the figures if they observed signal of centromeres at both sides of rupture, therefore, so far, their results suggest that breaks do not occur at alpha-satellite regions.
- 9) Model: The authors proposed that illegitimate loading of PICH/BLM promotes centromere disintegration. To claim this, they need to bring BLM artificially at the centromeres (using Cas9 or CENP-B boxes) to bypass Plk1 activity and test if this artificial tethering can promote the same phenotypes as observed in B12536.

Minor points:

- 1) The authors could even improve their figure schematics to indicate for each representative image (Figure 1e-g, Figure 2a and e...) which condition they represented. This will help the readers.
- 2) It has been described that UFBs can arise from different origins, including centromeres and telomeres (Chan and Hickson, 2011). Moreover, BLM is involved in the processing of telomeric regions (Sobinoff et al., 2017; Pan et al., 2017; Drosopoulos et al., 2015...). Based on the similarity between UFBs and PITs and because telomeric regions are also repetitive sequences, like the centromeric ones, it would be of interest to discuss why PITs do not originate from telomeric regions.
- 3) In figure 1f, g and h, a quantification of the colocalization of PICH / BLM and RPA on PITs should be presented. The same applies to figures 2a (no control is quantified), 3f and sup fig. 1g
- 4) The authors used several times the term "normal" to describe cell lines. What is the exact definition of the term "normal"? Non-transformed cell lines? I suggest rephrasing.
- 6) In the introduction, the authors stated: "its precise molecular action in resolving the intertwining chromatids remains largely unclear". As many important studies were performed to address this question, the authors should rephrase this sentence to take into consideration all the previous works.
- 7) The authors conclude that premature sister separation cannot be the cause of the defects that they observed following PLK1 inhibition measured by chromosome spreads. However, as they did not see separation even following sgo1 inhibition (see sup figure 2b), it is better to stain for sgo1 and quantify its intensity and localization before reaching this conclusion.
- 8) Page 5 (top), the authors concluded that: "this led us to speculate that the loss of chromosome alignment may not be simply attributed to the unstable KT-MT attachment as previously thought". They need to add "in the absence of active plk1"
- 9) Paragraph titles all along their results section would strongly help the reader.
- 10) It is unclear why the authors measured the efficiency of BLM depletion by siRNA as number of dots and not signal intensity measured by immuno-blot or RT-qPCR.

Reply to reviewers' comments:

We thank all three reviewers for their efforts and very constructive comments and useful suggestions for our study. We now provide additional data and explanation to support our study, whilst also amending our manuscript accordingly. We would like to kindly recommend the reviewers to refer to our figures displayed on a computer screen when reviewing the new version, as it provides the best clarity of our imaging results.

Reviewer #1 (Remarks to the Author):

The Plk1 cell cycle kinase has a number of important roles during cell division including the control of mitotic entry and bipolar spindle formation, the promotion of proper microtubule-kinetochore attachments and the orchestration of cytokinesis. Loss of Plk1 activity or the PLK1 molecule altogether results in spindle checkpoint mediated cell cycle arrest with monopolar spindle. Jones et al., investigate the role of PLK1 in promoting metaphase by inhibiting Plk1 activity using the well-characterised small molecule inhibitor BI2536. In contrast to previous reports they find that an aligned metaphase state is actually transiently achieved – at least in untransformed cells – but is then lost. Careful investigation of the Plk1-inhibitor treated cells revealed a loss of centromere integrity with a concomitant occurrence of PLK1-, PICH- and BLM-coated ultrafine DNA threads in the centromeric region (referred to as pre-anaphase inter-chromatin DNA threads (PITs)). In further experiments the authors report that this destabilization of the centromeric DNA leads to a splitting of chromosomes at the centromere into p- and q-arms. The authors refer to this event as “centromere dislocation”. Both the formation of the abnormal chromosome threads and the centromere splitting are dependent on the presence of PICH and BLM. BLM appears to be the key enzymatic activity for this event, whereas PICH seems to be mainly required to localize BLM. Altogether, the authors suggest that one crucial function for Plk1 during mitosis is to prevent untimely unravelling of the centromeric DNA due to uncontrolled BLM activity.

This study reports a very interesting novel aspect of PLK1 function during mammalian mitosis which will be of interest to a wide readership. The idea that PLK1 activity is required during mitosis to prevent unravelling of the centromere is particularly interesting. The experiments are conducted to a very high standard and the results are generally convincing and well presented.

We thank the reviewer for finding our study novel, of a very high standard and of interest to the wide scientific community.

The major shortcoming of this manuscript is that it is mainly descriptive and does not give any insight into how Plk1 may be exerting this novel, centromere-protective function.

In the current study, we have reported a number of novel mitotic phenomena relating to centromere maintenance by PLK1. We also identify the critical factors involved in this (PICH/BLM complex and bipolar spindle forces) and demonstrate how they mechanistically induce centromere disintegration, namely by the co-action of DNA unwinding and bipolar spindle pulling at centromere chromatin. Importantly, this provides a new explanation on how chromosome misalignment arises when PLK1 function is impaired. We agree with the reviewer that it would be interesting to

understand how PLK1 exerts this centromere-protective function(s), but we think that it should be subjected to future research, because it requires more comprehensive investigations.

I don't feel that this should preclude publication of these interesting observations, but the manuscript would certainly be improved by checking the obvious possibility that BLM is a target of Plk1.

We thank the reviewer for supporting the publication of our important findings.

Is BLM phosphorylated by Plk1? If so, could this modulate BLM activity? Minimally, an experiment addressing whether BLM is a Plk1 target or not (this could be done by Western blotting of BLM in extracts treated with vehicle control or BI2536) would nicely round off the manuscript.

BLM interacts with PLK1 and is hyper-phosphorylated during mitosis (Dutertre et al., *Oncogene*, 2000; Leng et al., *PNAS*, 2006). It is reasonable to think that BLM is a potential target of PLK1 and the hyperphosphorylation may regulate its function in mitosis. We now, as suggested by the reviewer, provide immunoblotting results showing that BLM as well as PICH are hyper-phosphorylated in mitosis in a PLK1-dependent manner (New Supplementary Fig. 14). Whether the PLK1-dependent phosphorylation modulates BLM (and PICH) interaction/recruitment/activity remains unclear and requires more careful, thorough investigations. For examples, mapping all phosphorylation sites and generating mutants for further analysis. Therefore, we think it should be considered beyond the scope of the current study.

The other slightly puzzling issue is that BI2536 has been used on cells for over a decade now, and the phenotype that Jones et al. describe has so far apparently been overlooked. Is this because most of the work on Plk1 has been conducted in HeLa cells? Do HeLa cells not develop the dramatic centromere splitting that Jones et al. see in RPE1 cells or are other defects, such as the bipolar spindle collapse, predominant in HeLa cells and mask the centromere splitting? It would be helpful to have a side-by-side comparison for some of the key experiments for HeLa and RPE1 cells (e.g. the live cell imaging).

We are glad that the reviewer pointed out that many previous studies use HeLa cancer cells as their studying model. We have now performed analysis on HeLa cells. Like other examined cell lines, HeLa cells also exhibit centromere disintegration after PLK1 inhibition. However, the frequency is lower (~21%) than in RPE1 cells (New Supplementary Fig. 4). Live-cell imaging analysis shows that HeLa cells are relatively poor at progressing into metaphase-like stage, or establishing chromosome biorientation under BI2536 treatment (New Supplementary Fig. 4). The reason is not clear but we agree with the reviewer that it is very likely that the bipolar spindle connection in HeLa cells is more vulnerable to the loss of PLK1, which therefore reduces the chance of centromere rupture.

Lastly, I find the terminology of “centromere delocalization” not very accurate. As far as I can see the centromere is split and not delocalized, so the correct description should be “centromere splitting” or “chromosome splitting”. Otherwise it’s very confusing for the reader.

We thank the reviewer for the suggestion. First, we would like to clarify that we use the term “dislocation”, not “delocalisation”. We use “dislocation” because the phenomenon of centromere separation looks similar to “a joint dislocation or luxation”. We agree that “splitting” is also a helpful terminology to describe the centromere damage. We now also use “chromosome-arm splitting/separation” for further clarification. However, we keep the term “dislocation” because we feel it is appropriate to describe this very specific mitotic centromere breakage. Secondly, we want to avoid confusion of “centromere splitting” with “sister-centromere separation” that the latter occurs in anaphase cells.

The same goes for “PIT”. With that name one automatically thinks of clathrin-coated pits and membrane trafficking. When reading the manuscript I had to go back several times to remind myself what was meant by this term. I am not sure why these structures have to be referred to by an acronym – “abnormal DNA bridges” seems perfectly adequate.

We thank the reviewer for the comment. We now remove the acronym “PIT” and use the term “centromere DNA threads” instead. We prefer not to use the term “abnormal DNA bridges” because in general, DNA bridges refer to DNA/chromatin intertwining structures linking between the separating sister chromatids, or fusion chromosomes in anaphase. We do not want readers to confuse centromere DNA threads with DNA bridge structures. The centromere DNA threads reported here are a very distinct chromosome deformation caused by the decompaction/decondensation along the centromere axis.

Reviewer #2 (Remarks to the Author):

Chan and co-workers report an analysis of the effects of perturbing PLK1 on centromere integrity. They reveal new phenotypes arising following PLK1 inactivation with chemical inhibitors. Some of these effects are apparently associated with an aberrant activity of the Bloom helicase. Overall, the work is conducted to a high standard, the subject matter is of significant interest to cell biologists, and the data are generally convincing.

We thank the reviewer for finding our work of a high standard, convincing and of significant interest to cell biologists.

However, there are some weaknesses and the study as it stands fails to establish causative connections between key events.

We have now provided additional data and clarification to support our key findings (please see responses below). Our study reports an unexpected consequence of the loss of PLK1 in mitosis, namely centromere disintegration. We further demonstrate that mechanistically, it is mediated through an aberrant unwinding of centromeric DNA by BLM helicase. How PLK1 suppresses centromere disintegration, and whether it is via control of the PICH/BLM complex, or by ensuring normal centromere configuration remains an open question. It is also possible that both or more processes are involved. In the revised manuscript we believe we provide a substantial amount of compelling evidence to demonstrate that PLK1 is vital for centromere integrity protection – a new function in mitosis.

Specific remarks:

1. I found the tone of the text far too dogmatic (things are stated as being 'proved' etc) and in places seriously overstated. This begins with the title, but there are also a lot of examples in the body of the manuscript. In particular, the claim in the title that BLM mediates dechromatinization is misleading in my view and should be dropped.

We thank the reviewer for the comments. We have amended and toned down the text accordingly. We think that one of the reasons the text may have come across as too dogmatic could be because we have not explained our data in detail (e.g. regarding to points 9 & 10). We now include additional data and clarification to support our claims.

As requested by the reviewer, we have changed the sentences using the word “prove”.

1. “To prove this, we examined mitotic spread chromosomes...” into “To determine if this was caused by chromatin damage, we”.

2. “Using multi-color FISH karyotyping, we further proved that all” into “...we further validated that PLK1.....”.

3. “To further prove that BLM and/or the PICH complex....” into “Next, we examined whether centromere disintegration is mediated by BLM and PICH.”.

Moreover, we have taken the reviewer's suggestion and dropped the word "dechromatinisation" and changed the title.

2. The summary states that BLM is 'illegitimately recruited'. My understanding from the literature is that BLM is found at centromeres anyway. What is the evidence in the current study for it being an illegitimate event?

Rouzeau et al have reported that BLM localises at centromeres. This was shown in late G2/prophase stages of Bloom's syndrome fibroblasts expressing an ectopic GFP-BLM, and on cytopun mitotic chromosomes of HeLa cells (Rouzeau et al., PloS One, 2012). Unfortunately, the centromere localisation of BLM has not been clearly shown in intact mitotic cells after nuclear envelope breakdown (e.g. from prometaphase to anaphase). Interestingly, studies from the same group also report that BLM is hyper-phosphorylated and excluded from chromatin during mitosis (Dutertre et al., Oncogene, 2000; Duetrtre et al., JBC, 2002). According to these findings, there may be another mechanism to keep a very small amount of BLM at centromeres, a level of which is probably under the usual detection limits.

In the current study, BLM signals are poorly or mostly not detected at centromeres in intact mitotic RPE1 cells under normal conditions. This could be again explained by a quantity of BLM that is below the detection limit of the immunostaining. However, upon PLK1 inhibition, strong BLM foci are readily detected underneath kinetochores. Because of the poor/undetectable signals of BLM in control cells, we thus use the term "illegitimate recruitment". However, we understand the reviewer's concern based on the previous report, so we now change "illegitimate" into "aberrant" recruitment.

3. A fundamental issue with studies of this sort where a kinase that plays multiple roles in a process is inhibited, is that it is very hard to dissect whether the apparent phenotypes are primary, secondary or tertiary effects of loss of the function of the enzyme. The structure of the kinetochore or centromere might be fundamentally different in the inhibitor treated cells, which then means that structures accumulate that don't exist in control cells. Hence, everything downstream of that becomes artefactual and doesn't reflect an event that would ever occur in 'real life'.

We fully understand the reviewer's concern. It remains unclear whether the destruction of centromeres is a primary effect because of mis-regulation of PICH and/or BLM in mitosis, or if it is a secondary effect triggered by potential malformation of centromere chromatin after PLK1 inhibition. It is also possible that both processes are involved. However, our data does seem to rule out the possibility that it is caused by problems in the initial formation of centromeres. We show that even a short inhibition of PLK1 (30min of BI2536 incubation) in "mature" mitotic cells; namely those cells that have gone through chromosome condensation and reached metaphase with active PLK1, it still triggers centromere disintegration (Fig. 8a, 8b). This result implies that cells seem to fail to maintain or protect centromeres without PLK1, rather than because of an establishment problem. This, however, still does not dissect the primary and the secondary effects. Interestingly, we also show that in the absence of PLK1 activity,

centromeres remain competent for kinetochore assembly and chromosome biorientation, as long as PICH or BLM is depleted (Fig.8c). Thus, if a structural defect does exist, it does not seem to largely influence centromere and kinetochore functions, but we agree that it may still lead to mis-activation of the PICH/BLM complex.

However, as we and others show that both PICH and BLM are substrates of PLK1 (New Supplementary Fig. 14), it is also reasonable to speculate that PICH/BLM activities are under PLK1-dependent regulation during mitosis. We agree with the reviewer, given that PLK1 has a number of substrates, we think it more likely that centromere disintegration is a combined consequence of the loss of centromere maintenance and mis-activation of the PICH/BLM complex. This thought is based on our and other previous studies that impairing chromosome condensation by condensin depletion alone does not seem sufficient to trigger severe centromere rupture as described here. More thorough studies will need to be conducted to dissect the different possibilities, and we think they should be subject to future investigation. No matter what the primary cause is, we think our data has demonstrated that PLK1 participates in a centromere maintenance pathway that has never been thought before.

With regret we do not agree with the reviewer's comment of '*...Hence, everything downstream of that becomes artefactual and doesn't reflect an event that would ever occur in 'real life'...*'. What we show here is a 'real biological consequence' of losing PLK1 functions in mitosis. We show that PLK1 plays a vital role in preventing this devastating centromere damage from happening during mitotic progression; thus, it reflects a physiological function in centromere maintenance. Our study provides an important starting point to explore this new pathway/concept of centromere maintenance. Importantly, apart from the common belief of KT-MT destabilisation, we provides a new mechanistic explanation on how chromosome misalignment can arise in PLK1-defective cells.

Some more balanced discussion reflecting the inability to distinguish primary from secondary events would be appropriate.

We have taken the reviewer's suggestion and incorporated different possibilities on how centromere disintegration may arise.

4. On a related note, the authors failed to demonstrate any functional connection between PLK1 and BLM. Without that, it is hard to be sure that they are not studying phenomenology that has no relation to cell physiology. It is important to make that connection and to determine how BLM is inhibited by PLK1.

The cause(s) of centromere disintegration may be determined by single or combined factors, such as the loss of centromere maintenance and/or the mis-regulation of PICH and/or BLM activities in mitosis. These could be due to the lack of phosphorylation on

condensin, histones, the PICH/BLM complex, or other unknown factors. We have identified that the helicase activity of BLM is crucial in centromere disintegration and have also shown that both PICH and BLM are substrates of PLK1 (new Supplementary Fig. 14). However, whether the lack of hyper-phosphorylation on BLM alone is the sole determining factor for centromere disintegration remains uncertain.

We totally agree with the reviewer that a direct functional connection between PLK1 and BLM, if exists, will provide more mechanistic insights. But given the multifaceted role of PLK1 and its substrates in mitosis, we believe it probably involves multiple factors such as PICH and/or condensin phosphorylation. Therefore, focusing merely on PLK1 and BLM's relationship may not truly reflect the whole picture, so a more comprehensive study will be required in the future. Importantly, our study has provided an alternative mechanism to explain how chromosome misalignment can arise in PLK1-inactive cells. We believe this finding would be very helpful for scientists to further understand the biological functions of PLK1 and the role of UFB-binding complex in mitosis. Therefore, we think that no matter if PLK1 directly or indirectly modulates BLM activity, our study still highlights the new important role of PLK1 in centromere integrity maintenance.

5. Is there a need to call the claimed 'new' DNA structure by a specific name? The PITs show the same characteristics as UFBs, which are also visible at centromeres before anaphase (shown by the Nigg group). What is the evidence that PITs are anything fundamentally different from UFBs?

We would like to emphasise that UFBs are generally used to describe DNA linkage structures that intertwine 'between sister chromatids'. They could be unresolved dsDNA catenanes, under-replication of DNA or homologous recombination structures. It is correct that a very short UFB/PICH focus can be observed at centromeres in late metaphase cells, but they are located in between separating sister centromeres (Please see Supplementary Fig. 10a, in untreated metaphase - yellow arrows). Presumably, it represents the sister DNA catenanes.

The centromeric DNA thread described in the current study is the centromere chromatin that connects the short and long arms of chromosome. Their formation is caused by the decompaction of the chromatin along the axis, which seems not related to DNA intertwinements between sister centromeres. Moreover, we detect condensin on the DNA threads, which is virtually undetectable on anaphase UFBs (Supplementary Fig 7, Chan et al., EMBO J, 2007). Therefore, UFBs and the centromere DNA threads are fundamentally caused by different types of DNA structures.

6. The claim that spindle pulling forces cause the stretching and other distortions of the DNA is not fully demonstrated by the data. At best, nocodazole partially suppresses the effects of PLK1 inhibition. For example, in Figure 2b, statistical analysis of Noc versus BI+Noc should be performed. They look very different to me. If so, that would change the strength of any conclusion.

Disruption of spindle formation by nocodazole significantly reduces centromere dislocation ($p=0.019$) (moved to Fig.3b) and chromosome fragmentation (from 58 to 48 chromatin, Supplementary Fig. 6c). We now provide the statistical analysis between NOC and BI+NOC ($p=0.0744$) (Fig. 3b), which it is not significant. Clearly, the bipolar spindle pulling forces play a significant role in centromere disintegration. However, we agree with the reviewer that there are some levels of centromere breakage and partial centromere splitting even after nocodazole treatment. This may imply other factors such as chromosome arm hyper-compactation may influence centromere integrity. Alternatively, it could be a technical issue. As centromere DNAs have been unwound, it may be predisposed to split during the metaphase spread preparation.

7. Could the authors comment on how spindle pulling could cause longitudinal stretching of the chromosome?

We amend our text as we think the term 'longitudinal stretching' may be mis-leading to readers. Centromere disintegration occurs after chromosome biorientation/metaphase alignment. Besides, we now also show that monopolar attachment, generated by using monastrol, is not sufficient to induce centromere rupture (Supplementary Fig. 8). This suggests that a tension across the centromere, likely from bipolar spindle attachment, is required. Because sister chromatid cohesion remains present, simultaneous unwinding and pulling of centromeric DNA on each side leads to chromatin decompactation and results in the separation of the short and long arms (please see our model in Fig. 9 for details). This breakage outcome probably generates an illusion of a longitudinal stretching.

8. The authors rightly state that PLK1 is not needed for PICH to localize to centromeres. However, the Nigg group showed that PLK1 depended on PICH. If true that would really complicate the picture. I feel that this issue needs to be investigated.

Two studies from the Nigg group show that the localisation of both PICH and PLK1 is independent of each other at centromeres (please see figures on the next page). However, the loss of PLK1 can lead to additional loading of PICH to chromosomal arms (Figure 5A & D, Santamaria et al, MBC, 2007; Figure 2A, Baumann et al, Cell, 2007). These data are consistent with our finding that PLK1 and its activity is not required for PICH to localise to centromeres (Supplementary Figs. 9e, 10a-d). On the other hand, as far as we know, there is no available study showing that the activity or function of PLK1 requires PICH. But if it does, depletion of PICH should cause some forms of PLK1-deficient phenotypes such as mitotic arrest. However, that is not the case.

Figure 5A & D, Santamaria et al, MBC, 2007

9. RPA forms a complex with BLM and therefore using RPA to define ssDNA might be complicated by the fact that it might be recruited to centromeres through binding BLM. A second method to detect ssDNA would be valuable (native BrdU?).

We thank the reviewer for raising this important point. We should have explained this in our first submission. RPA interacts with BLM *in vitro* (Brosh Jr et al., JBC, 2000). If the presence of RPA at centromeres (and on DNA threads) is because of the physical interaction with BLM, two important outcomes should be observed. First, RPA and BLM staining should display a high co-localisation pattern. Secondly, the presence of RPA should be independent of BLM's helicase activity. However, our data shows that both predictions are not the case. In fact, we find that although PICH, BLM and RPA proteins are detected along the same DNA threads, RPA localisation does not always follow the PICH/BLM signals. In addition to regions positive for PICH/BLM, RPA also binds to those regions with no or weak PICH/BLM signals (Fig. 2c, 2d). We have used different antibodies against BLM and different RPA subunits, and obtain the same profile. Moreover, this result is consistent with our finding on anaphase UFBs (Fig. 3, Fernández-Casañas & Chan, Genes 2018) and other using single-molecule analysis (Fig. 5b, Sarlos et al., NSMB, 2018)(Please see figures on the next page). Most importantly, it has been shown that the helicase activity, but not BLM protein itself, is required for RPA loading on UFBs (Fig. 6a, Chan et al., Nat Cell Bio., 2018).

Fig. 3, Fernández-Casañas & Chan, Genes 2018

Fig. 5b, Sarlos et al., NSMB, 2018

Fig. 6a, Chan et al., Nat Cell Bio., 2018

We also provide new data to show that the loading of RPA at centromeres is dependent on the DNA unwinding activity of BLM, and not its physical occupancy. We show that in the absence of PLK1 activity, the helicase-dead BLM is aberrantly recruited to centromeres but fails to induce RPA foci (New Fig. 6d). This strongly indicates that there is a lack of ssDNA formation. Thus, RPA is a genuine ssDNA marker, at least in the current study. Crucially, the helicase-dead BLM also fails to induce centromere disintegration.

We thank the reviewer for the suggestion of using BrdU staining. Native BrdU immunostaining can reveal single-stranded DNA structures. However, it only works if the BrdU epitope is not sterically hindered (e.g. on a naked single-stranded DNA). We have tried native BrdU staining but the signal is undetectable. Given that most of the DNA thread-like structures are heavily decorated by RPA, a very plausible explanation is that the RPA molecules block the access of the BrdU antibody. This problem also happened in the detection of ssDNA on UFBs in our previous study (Chan et al., EMBO J 2007). The efficiency of BrdU staining on DNA bridging structures, even under denaturation conditions, is very inefficient. Nevertheless, based on the facts that RPA localises differently to the PICH/BLM complex along DNA threads, and its binding requires BLM's helicase activity, this would strongly suggest there is illegitimate ssDNA formation at centromeres and centromere DNA threads after PLK1 inhibition.

10. The experiments with the Q672R mutant of BLM are compromised by the fact that the mutant protein is expressed at a significantly lower level than the control cells. This is fine if the mutant complements the defect, but it doesn't in this case. A better pair of matched clones needs to be analyzed.

We agree that the protein level of GFP-BLM(Q672R) mutant is lower compared to the GFP-WT. However, the level of GFP-Q672R is indeed very similar to that of endogenous BLM, whereas the GFP-WT is overexpressed (Supplementary Fig. 13b). As stated in the method, these GFP-BLM cell lines are polyclonal, therefore we cannot isolate clones expressing a higher level of the mutant GFP-Q672R.

However, to address the reviewer's concern, instead the Q672R, we re-sorted the wildtype GFP-BLM cells and obtained a new polyclonal cell line expressing a much

lower level of wildtype GFP-BLM protein – significantly lower than GFP-Q672R and endogenous BLM (New Supplementary Fig. 13c). We have now repeated DNA thread analysis and found that despite the low protein abundance, the wildtype GFP-BLM is still able to trigger centromere disintegration after PLK1 inhibition (New Supplementary Fig. 13d). Therefore, the failure of GFP-Q672R to induce centromere disintegration is not because of protein inadequacy, but instead because of the lack of DNA unwinding activity.

In addition, we also measured the centromeric RPA foci formation/intensity at kinetochores in GFP-WT and GFP-Q672R cells. We found that irrespective of the protein levels, the Q672R mutant, even binding higher at some kinetochores, always fails to generate RPA foci (New Fig. 6d). Therefore, we conclude that ssDNA formation at centromeres and the subsequent centromere disintegration requires BLM-mediated centromere unwinding.

Reviewer #3 (Remarks to the Author):

The Polo like kinase 1 (Plk1) is one of the most studied kinase mainly required to regulate chromosome segregation during mitosis. Inhibition of Plk1 activity is known to affect phosphorylation of several key components of the mitotic network leading to mitotic arrest and cell death.

In this manuscript, the authors described the consequences of PLK1 inhibition on metaphase chromosomes. They first observed that PLK1 inhibition in metaphase leads to the formation of a new type of DNA structure that they called ‘pre-anaphase interchromatin DNA threads’ (PITs). This structure somehow resembles the previously identified Ultrafine anaphase bridges (UFBs). They proposed that these structures result from the illegitimate recruitment of BLM at the centromeres under conditions of PLK1 inhibition. Due to the mitotic spindle forces, these bridges ultimately fail to withstand spindle tension, resulting in whole chromosome arms splitting.

The authors performed a large amount of work and described several and novel outstanding phenotypes that are certainly of interest to the community and the reader of Nature Communications.

We thank the reviewer for finding our results novel and of interest to the scientific community.

Nevertheless, the interpretation of some of these observations remain poorly established and some key experiments are missing to test their model. In addition, several main points are contradictory with the literature and have to be addressed before publication in Nature Communications.

We now provide additional data and explanation to address the reviewer’s concerns.

Major points:

1) BLM protein has been shown to be recruited from G2 to mitosis at the centromere leading to changes in centromeric structure together with topoisomerase IIa. Their cooperation is required to resolve centromeric UFB. BLM depletion is associated with several mitotic abnormalities, including UFB and chromatin bridge formation as described by the same corresponding author some years ago (Chan et al., 2007, Baumann et al., 2007, Chan and Hickson, 2011). So, it is well established that BLM activity is necessary to resolve abnormal DNA structures during mitosis, thus preventing genetic instability.

Here, the authors proposed that, during an unperturbed mitosis, BLM activity is inhibited by PLK1 in order to maintain centromeric integrity, therefore preventing PITs formation (a DNA structure very similar to the UFBs).

We would like to clarify that the nature/origin of the centromeric DNA threads (PITs) identified here is fundamentally very different from anaphase UFBs, although they are both recognised by the PICH/BLM complex. The UFB structures are generally referred to as DNA linkages intertwining in between “sister chromatids”. They can be caused by double-stranded DNA catenanes, under-replicated DNA structures and homologous recombination structures. In the current study, the centromeric DNA threads represent a decompaction of chromatin from the core centromere, which

arises between the short and long arms of chromosomes, not between the sister centromeres.

Their data imply that BLM depletion rescues drastic chromosomal defects as PITs formation and loss of centromere integrity, which is very surprising based on the previously described role of BLM. The authors should explain this discrepancy with the literature. Indeed, it is difficult to imagine why BLM is recruited at the centromere during mitosis if its function leads to such catastrophic events in the absence of PLK1 activity.

BLM-deficient cells exhibit increased chromosome segregation defects such as anaphase DNA bridge formation. Because of the UFB-binding property, it is generally believed that BLM participates in resolving DNA bridge structures during anaphase. Our current study shows that BLM can also decompact (or resolve the centromere chromatin) when PLK1 activity is disturbed.

Rouzeau et al (PloS One, 2012) observed BLM foci at centromeres in late G2/prophase cells of Bloom's syndrome fibroblasts expressing an ectopic GFP-BLM, and on cytopun mitotic chromosomes of HeLa cells. They also reported that there is a change in centromere structure (volume) in BLM-deficient cells. Thus, it leads to a proposal that BLM may function at centromeres even before chromosome segregation. Unfortunately, centromeric BLM foci are rarely detected in intact prometaphase and metaphase cells. It could be due to detection limits. However, it is also unclear whether the centromeric structure change observed in BS cells is caused by (i) the lack of the proposed centromere function in early mitosis, (ii) abnormal DNA replication/repair processes during S phase, or (iii) because of accumulating defects in centromere segregation in previous rounds of anaphase. The same group also reports that BLM is hyper-phosphorylated and excluded from chromosomes during mitosis (Dutertre et al., Oncogene, 2000; Duetrtre et al., JBC, 2002). These results seem to suggest that there is a regulation on BLM's localisation and/or activity in early mitosis. As we and other groups show that BLM (and PICH) are substrates of PLK1, it is reasonable to think that PLK1 is one of their regulators in mitosis.

In our current study, we demonstrate that PLK1 has a vital function in centromere protection. Our data shows that the lack of PLK1 activity triggers illegitimate unwinding of centromeric DNA by BLM before chromosome segregation. This results in centromere decompaction and the separation of the long and short chromosome arms. Conceivably, if a similar decompaction reaction occurs at sister chromatids that are intertwined by DNA entanglements, BLM may be able to relieve the constraints and facilitate the chromatids separation in anaphase. If correct, this activity could explain and fit nicely to what other literatures suggest the role of BLM in chromosome segregation. However, this BLM-mediated decompaction activity must need to be under a tight control, presumably by PLK1; otherwise, it could cause catastrophic centromere damage. Alternatively, the lack of PLK1 activity may lead to a change in centromere structure which mis-activates the PICH/BLM complex in early mitosis. Thus, our study suggests that there may be a mis-regulation/mis-activation of the

PICH/BLM pathway when PLK1 activity is compromised. Therefore, our results are not contradictory to the suggested role of BLM in segregation when it is under a proper control. Instead, our finding provides a possible hint on how BLM may facilitate the separation of intertwined sister chromatids in anaphase.

2) Most of the experiments have been done upon several synchronization events rendering the interpretation of the results difficult. For example, abnormal nucleotide pool concentration is known to induce replication stress and so thymidine treatment could explain some phenotypes observed by the authors. It is not clear to this reviewer why the authors had to synchronize cells at all as they specifically look at cells during mitosis, so they could perform all their experiments in unsynchronized and unperturbed cells.

RPE1 cells are hTERT-immortalised diploid epithelium cells, which have a low mitotic population. Using the single thymidine block-release approach first helps to enrich G2/M populations and secondly, minimises the variation of treatment time due to asynchronous entry of mitosis. Under our experimental procedure, after five or six hours post G1/S release, the majority of cells reach G2/M (Supplementary Fig. 1b), where we treated with BI2536 for another 2 to 3 hours (mostly 2hrs). This synchronisation/treatment approach also helps us to prepare mitotic spreads without using spindle poisons (e.g. Colchicine and nocodazole). Since we found that bipolar spindle tension is crucial factor for centromere disintegration, it would be very inappropriate to use spindle poisons for mitotic chromosome preparation. We also believe this is the reason why the important phenotype of centromere disintegration has not previously been reported. Unfortunately, without using any synchronisation methods, it is almost impossible to obtain adequate numbers of chromosome spreads for analysis.

For DNA thread analysis, we have now shown that BI2536 treatment (for 1hour) also triggers centromere DNA threads in unperturbed asynchronous RPE1 cells (New Supplementary Fig.3d). However, because of the asynchronicity, we notice that some mitotic cells that are already in late prometaphase and metaphase, when the drug is added, escape into anaphase. So, it is less efficient to see centromere disintegration from an asynchronous population compared to synchronous cells. Nevertheless, our data clearly shows that centromere disintegration is not caused because of thymidine pre-treatment.

The authors have to show that centromeric dechromatinization and disintegration arise in unsynchronized cells as well (and not with 16hr of CDK1 inhibitor as they did in sup figure 3c) eliminating thus a potential role of the synchronization in the phenotypes described in this paper.

Initially, we had the same concern as the reviewer on using thymidine, and that is why we also use a CDK1 inhibitor to arrest cells in G2, and then release them into mitosis in the presence of BI2536. It shows the same centromere rupture phenotype. Now, we show the centromere damage can be induced in asynchronous RPE1 cells after 1hr

BI2536 treatment (New Supplementary Fig.3d). Therefore, we believe that centromere disintegration is not caused by any synchronisation procedures.

More importantly, most of the phenotypes described in this paper result from PLK1 inhibition during 8 hours (cf Figure 1a), whereas mitotic duration in RPE cells is estimated at 50 minutes. Thus, in this condition the mitotic duration is extended by 8 times rendering it difficult to appreciate the biological significance of the subsequent phenotypes. The authors have to discuss this point.

We apologise that we did not describe our experimental procedures clearly and that led to confusion. We now amend our labelling (e.g. 8hr to t=8hr). Briefly, drugs are added at 5 or 6 hours post the G1/S release. After 2-3 hours incubation (t=8hr, 8hrs post thymidine release), cells were harvested for analysis. So, the G2/M populations are only treated for 2-3hrs and not for 8 hours as the reviewer interprets. We also want to emphasise that at ~6 hours post G1/S release, most cells are still in an interphase stage, namely late S or G2. Therefore, even though we apply BI2536 for 2 hours before analysis, most of the subsequent emerging mitotic cells are in fact inhibited by the drug for less than 2 hours.

On this regard, it is unclear which control the authors used. Which phase is compared here in DMSO (e.g. Figure 1e, but this is true for many of all other figures)? As time in mitosis is very different between cells with PLK1 inhibition and cells treated with DMSO, it is better to have a similar control of mitotic phase, for example by inhibiting CENP-E.

Using the same synchronisation period, at t=8hr (8hrs post G1/S release), DMSO control shows an enrichment of mitotic populations, mainly in normal prometaphase/metaphase. We call them a 'pre-anaphase population' and use them as a control comparison. We believe this is a right control population. Additionally, we also perform comparisons of BI2536 vs nocodazole+BI2536 and BI2536 vs monastrol+BI2536 (they are all arrested in mitosis) and show that bipolar spindle pulling tension is required for centromere DNA thread formation (New Supplementary Fig. 8).

3) Topoisomerase IIa has been extensively described to promote the resolution of centromeric DNA structures during mitosis (Porter and Farr, 2004), in association with PICH and BLM (Nielsen et al., 2015; Rouzeau et al., 2012). Given the importance of this protein in the processing of centromeric DNA, the authors have to show whether topo IIa activity is involved in PIT formation.

TOP2A is crucial to resolve dsDNA catenanes. Loss of TOP2A activity increases anaphase UFBs, presumably because of unresolved dsDNA catenanes that link the separating sister chromatids. The centromere DNA thread structure identified here is a fundamentally different structure. It is unlikely related to dsDNA catenations between sister centromeres, as it is the decompacted chromatin axis along the centromere. We are not sure why TOP2A activity might be involved in centromere DNA thread formation as suggested by the reviewer. Nevertheless, we have treated cells with a

high dose of TOP2 inhibitor (ICRF193) during the BI2536 treatment, but it does not prevent DNA thread formation. As we think this result is not crucial for our interpretation, we do not include it in the manuscript.

4) To test the implication of the spindle forces on PIT formation the authors used Nocodazole treatment that leads to microtubule depolymerization. This treatment can test a direct role of microtubules but cannot be used to directly analyze the implication of mitotic spindle forces. A better way to do that would be to use Eg5 inhibitor to inhibit outward forces or dynein inhibitor to inhibit inner forces. The authors should explore the role of the main drivers of mitotic spindle forces to exclude an implication of these forces in PIT formation.

We show that centromere DNA threads and subsequent centromere disintegration, appear after chromosome biorientation (Fig. 8a, b & Supplementary Fig. 1f) and that their formation is abolished by nocodazole treatment (Fig. 3b & Supplementary Fig.8). This implies that this process is very likely dependent on spindle pulling forces. We have taken the reviewer's suggestion to use the Eg5 inhibitor, monastrol, to further test if this is caused by spindle forces. As predicted, like nocodazole, monastrol also suppresses the formation of centromere DNA threads (New Supplementary Fig. 8). Therefore, this new data supports the claim that centromere disintegration is dependent on bipolar spindle tension.

5) The authors conclude that PITs correspond to DNA structures based on RPA staining (and presumably binding). This is indirect proof. The authors should use DNase treatment or BrdU incorporation (as previously done for UFB characterization (Chan et al., 2007)) to clearly established that PITs are DNA structures.

We thank the reviewer for the suggestion of using the DNase treatment and BrdU methods that we previously published (Chan et al., 2007). DNase treatment indeed is an indirect way to show DNA structures. This is because if the threadlike molecules are a form of cytoskeleton that builds on centromere chromatin, the digestion of the "DNA foundation" will inevitably destroy the staining. We have tried BrdU labelling, but unfortunately, BrdU signals are not detectable. I must emphasise that based on our experience, the BrdU method is very inefficient for DNA bridge labelling (Chan et al., 2007). As the centromere DNA threads are fully decorated by the PICH/BLM complex, RPA and even condensin, we speculate that the occupancy by these proteins may prevent the access of anti-BrdU antibody.

In spite of this, we strongly feel that during our current study we have provided ample evidence to demonstrate that the structure we observe is in fact a DNA structure. Over the last decade, studies have demonstrated that PICH (a DNA translocase) binds to ultrafine DNA bridges, stretched DNA (Baumann et al., Cell, 2007; Sarlos et al., NSMB, 2018) and processes triplex DNA (Biebricher et al., Mol Cell, 2013). The association of BLM helicase to UFBs is dependent on PICH (Ke et al, EMBO J, 2011). Crucially, the loading of RPA on UFBs requires BLM's DNA unwinding activity (Chan

et al., Nat Cell Bio., 2018), suggesting the requirement of ssDNA structures. All of these studies strongly indicate that the PICH/BLM complex specifically associates with and modifies DNA bridging structures. In addition, here, we also detect the presence of condensin (a complex for chromosome condensation) on centromere threads. Moreover, the threads are not co-localised with cytoskeleton microtubule structures (Supplementary Fig. 1h). But most importantly, our cytogenetic FISH analysis perfectly matches the cytological data showing that centromeres are torn apart after PLK1 inhibition. In some cases, we even observe an ultrafine DNA thread labelled by CEN FISH DNA probes (Fig. 3a, connecting arrows). This should be considered as direct proof of centromeric DNA structures. Considering all previous research and both our cytological and cytogenetic data, we think it is very fair to conclude that the centromere threads are DNA structures. On the other hand, it would be very difficult to conceive how all these DNA modifying proteins build a threadlike structure without the involvement of DNA molecules.

6) BLM protein mainly acts in a complex with Topo III / RMI1 / RMI2 forming the BTRR complex (Manthei and Keck, 2013). This complex is crucial to resolve linked DNA intermediates. More importantly, recently a new model implicating the BTRR complex in the resolution of DNA structure mimicking DNA threads (comparable to UFB or PIT structures) has been proposed (Sarlós et al., 2018). It will be of interest (but not absolutely required) if the authors could test and discuss if the BTRR complex is also involved in PIT formation/resolution.

We have provided a substantial amount of data and novel findings. We would like to keep our study focused on BLM-mediated centromere disintegration; whether TOP3A, RMI1 and 2, hRIF1 are involved or not does not seem to alter the main conclusion and it should be subject to future study.

7) Figure 2b, chromosomes look very weird and not convincing. All chromosome regions seem to be affected by the inhibition of PLK1, not only centromeres. Is this due to loss of PLK1 activity or prolonged mitosis or simply due to technical problems with this spread+FISH. As the chromosomes look normal in other experiments done w/o FISH, the last option seems to be the most likely.

We apologise we did not make clear that those are deconvolved high-resolution images (move to Fig. 3a & New Supplementary Fig. 7a). We now also provide the corresponding raw images for comparison (New Supplementary Fig. 7a, bottom panels). PLK1 inhibition makes chromosome arms shorter and thicker as mentioned in the results. Deconvolution helps to reveal better chromosomal details and signals. It makes some regions look like gaps, but this is probably some chromosome banding patterns. The chromatin arm regions are not broken or separated, except at the centromere in the BI2536-treated cells.

The authors should try to perform an immunofluorescence coupled with chromosome spread and stain for telomeres and centromeres to confirm their findings.

We thank the reviewer for the suggestion of using immunofluorescence staining on cytopun mitotic chromosomes. However, based on our experience, the centrifugal cytospinning forces can introduce artefactual chromosome breakages and structural damage.

In (clinical) cytogenetic analysis, mitotic spreads are prepared by using methanol:acetic acid fixation. Coupled with FISH, this method allows chromosome structures and rearrangements to be studied. This is considered as the 'gold standard' method, which is employed in our study. Mitotic chromosome spread with FISH provides two advantages; first, it directly labels and analyses the DNA regions of interest by using DNA FISH probes. Secondly, it minimises the introduction of artefactual chromosome breakage during chromosome spread preparation. Now we present additional data to show that virtually all chromosome breakages are found at the core centromere (new Fig. 3c & New Supplementary Fig. 7b). Together with centromeric DNA thread formation data and (m)FISH cytogenetic analysis, we believe all of our data clearly demonstrates that the chromosomal breakage occurs specifically at core centromeres.

8) To better understand if centromeres break at centromeres or at their proximity, the authors should measure CENP-A and CENP-B boxes (or CENP-B) amount. If the breaks are at centromeres, intensity of CENP-A/B will drop. If not, this would suggest that breaks do not occur at centromeres.

The question is whether the breaks occur at or outside core centromeres. We now provide additional images and data (Fig. 3c and Supplementary Fig. 7b) to show that almost all broken chromatin examined retains CEN FISH signals (total 524 chromatin; 99.6% CEN +ve). The only explanation is that the breakage occurs within centromeres/alpha-satellite regions.

We had shown the data of the occupancy of CENP-A and NUF2 at centromeres in pre- and post-metaphase collapse cells (now moved to Fig. 5d). The results show that most centromeres following metaphase collapse lose one of the two kinetochore complexes. However, this experiment was carried out to test if centromere disintegration might destroy kinetochore attachments. It cannot be used to determine if there is a centromere breakage because centromeres missing CENP-A/NUF2 signals could be because of failure in kinetochore assembly, not necessarily DNA break. Measuring CENP-B intensity, we think, is an indirect way to determine centromere breakage as a centromere break should not change the CENPB intensity, but rather reduce the area of CENP-B regions. The "gold standard" method, as we have employed, is to directly visualise the physical breakage at centromeres on mitotic spread chromosomes.

Another approach that the authors can use is to perform chromosome spreads using specific centromeric probes to see if they observe centromeric signal at both sides of a broken chromosome.

We thank the reviewer for the suggestion of using chromosome-specific centromere DNA probes. We have used a pan-centromere DNA probe to analyse CEN FISH signals on all broken chromosomes, and we find 99.6% of them retain CEN sequences (Fig. 3c and Supplementary Fig. 7b). Therefore, the only explanation is that the rupture occurs within the core centromere.

It is unclear from the figures if they observed signal of centromeres at both sides of rupture, therefore, so far, their results suggest that breaks do not occur at alpha-satellite regions.

Different centromeres have different lengths of alpha-satellite repeats. So, the CEN FISH signal intensities vary between centromeres. This may make some of the broken chromatin arms seem negative for CEN signals. We now provide new data and additional images with enhanced contrasts to show all broken chromatin retain CEN signals (Fig. 3c & Supplementary Fig. 7b). Generally, we avoid presenting images with over-contrast-enhancement because it saturates signals in other regions with strong staining. We think this is not appropriate. But we understand the reviewer's point, so we provide the contrast-enhanced images (Supplementary Fig. 7b). Evidently, CEN signals are observed in all intact and broken chromosomes.

On the other hand, we would like to mention that some CEN signals may not always be seen at the tip of the broken chromosome. This is not because the breakage is outside the core centromere. Instead, this is because the chromatin is in a 3D compacted structure, and at certain angles, it looks like the CEN signal is not present at the terminal end of the broken chromosome under a 2D image. This also applies to telomere FISH signals that are not necessarily seen at the very end of the chromosomes.

9) Model: The authors proposed that illegitimate loading of PICH/BLM promotes centromere disintegration. To claim this, they need to bring BLM artificially at the centromeres (using Cas9 or CENP-B boxes) to bypass Plk1 activity and test if this artificial tethering can promote the same phenotypes as observed in B12536.

We have completed the centromere tethering experiments by using CENP-B-BLM fusion protein. We did this in HeLa cells and RPE1 cells. However, transient over-expression of either wildtype or a CENPB-fusion BLM protein in RPE1 cells prevents cell cycle progression and reduces the mitotic population, which limits our analysis. However, we were able to track a number of transfected HeLa cells progressing into mitosis, but we did not detect any mitotic effects similar to PLK1 inhibition, e.g. metaphase collapse or arrest. Cells that over-express CENPB-GFP-BLM remain capable of progressing through anaphase (New Supplementary Fig. 15). The centromere tethering of BLM also does not induce RPA foci/ssDNA formation. This probably suggests that additional factors, e.g. centromere remodelling and/or

regulation of PICH and BLM may be also required, instead of a simple centromere over-loading. To avoid confusing readers by suggesting that the illegitimate loading of BLM is a main cause of centromere disintegration, we have also amended our model and discussion to include other possible factors.

Minor points:

1) The authors could even improve their figure schematics to indicate for each representative image (Figure 1e-g, Figure 2a and e...) which condition they represented. This will help the readers.

We thank the reviewer for the suggestion and we have amended the figures accordingly.

2) It has been described that UFBs can arise from different origins, including centromeres and telomeres (Chan and Hickson, 2011). Moreover, BLM is involved in the processing of telomeric regions (Sobinoff et al., 2017; Pan et al., 2017; Drosopoulos et al., 2015...). Based on the similarity between UFBs and PITs and because telomeric regions are also repetitive sequences, like the centromeric ones, it would be of interest to discuss why PITs do not originate from telomeric regions.

We would like to emphasise that the nature and origin of UFBs and centromere DNA threads (previously termed PITs) are fundamentally different. UFBs mostly represent the DNA intertwining structures that arise between sister chromatids. In principle, they can form anywhere along chromosomes; whereas centromere DNA threads are the decompacted chromatin of core centromere, caused by the co-action of DNA unwinding and bipolar spindle pulling. There is no spindle attachment and pulling (and PICH/BLM localisation) at telomeres, thus chromatin decompaction and DNA thread formation cannot occur at telomeres.

3) In figure 1f, g and h, a quantification of the colocalization of PICH / BLM and RPA on PITs should be presented.

(Fig. 1f moved to 2c) We have counted 88 DNA threads. As expected, all threads (100%) are positive for PICH, BLM and RPA. We also present consecutive z-stacking images. However, we would like to emphasise that RPA tends to bind to regions of DNA threads that show weak or no PICH/BLM staining. This indicates that the RPA loading, like on UFBs, is not simply a result of physical recruitment by BLM.

(Fig. 1g moved to 2d) We show the frequency of DNA threads originating from centromeres. The quantification was shown below the images (171/171 threads are centromere linked).

(Fig. 1h moved to Fig. 2e) An example to show that centromere DNA threads link between chromosomes that remain fully cohesed. It is used to support the claim that sister chromatid cohesion remains after PLK1 inhibition (no premature loss of cohesion in 90 spreads, Supplementary Fig. 2c).

The same applies to figures 2a (no control is quantified), 3f and sup fig. 1g

(Fig. 2a moved to Supplementary Fig. 5a) We show a DNA damage response as marked by gammaH2AX staining occurring around centromeres and mainly after but not before metaphase collapse. The control population is the one before the collapse, i.e. prophase and prometaphase. In wildtype cells, there are no collapsed cells to compare. Nevertheless, we performed gammaH2AX IF analysis on prophase and prometaphase in DMSO-treated cells, and as predicted, gammaH2AX signal is rarely or not found at (peri)centromeres (New Supplementary Fig. 5b). This is not because of staining failure as we can readily detect gammaH2AX signals at some telomeres, which has been reported before.

(Fig. 3f moved to Fig. 5a) This is a very rare event and found in less than 4% of centromeres in metaphase-like cells.

Supplementary Fig. 1g shows whether PLK1 is detectable on UFBs. We have looked at 64 PICH-stained UFBs and as expected, all (100%) are PLK1 positive.

4) The authors used several times the term “normal” to describe cell lines. What is the exact definition of the term “normal”? Non-transformed cell lines? I suggest rephrasing.

We now use the words “hTERT-immortalised diploid cells” and “primary cells”.

6) In the introduction, the authors stated: “its precise molecular action in resolving the intertwining chromatids remains largely unclear”. As many important studies were performed to address this question, the authors should rephrase this sentence to take into consideration all the previous works.

We have changed this sentence to “...precise molecular mechanism of UFB resolution is not yet fully understood”. We believe this is a fair statement because apart from the knowledge that UFBs are bound by the PICH/RIF1/BLM complex and RPA (which depends on BLM’s helicase activity), there is no direct evidence to show how mechanistically the intertwining DNA molecules are resolved during chromosome segregation. Although many studies show that defects in the UFB-binding complex increase anaphase UFBs, the facts that the BLM complex also has important roles in DNA replication/repair, the increase in UFB formation could be still a consequence of the accumulation of abnormal intertwining structures in S phase. On the other hand, defects in PICH also influences chromosome condensation, which could, in principle, increase UFB formation. Besides, there is no convincing explanation why most of the entire length of UFBs is covered by the UFB-complex, and why a long region is converted into ssDNA if there is only one or a few intertwining “junctions”. Whether the translocase and/or topoisomerase activity participate in UFB resolution remains unknown. What the UFB-complex is doing on UFBs, again is unclear. Because of all these questions, we think that the precise mechanism of the UFB-binding complex during chromosome segregation remains not understood.

7) The authors conclude that premature sister separation cannot be the cause of the defects that they observed following PLK1 inhibition measured by chromosome spreads. However, as they did not see

separation even following sgo1 inhibition (see sup figure 2b), it is better to stain for sgo1 and quantify its intensity and localization before reaching this conclusion.

We are not sure why the reviewer claims there is no premature sister chromatid separation in the Sgo1 knockdown cells (now moved to Supplementary Fig. 3a). We now provide an additional chromosome spread image and scanline measurements to show that sister chromatid cohesion is lost in Sgo1 RNAi cells.

8) Page 5 (top), the authors concluded that: "this led us to speculate that the loss of chromosome alignment may not be simply attributed to the unstable KT-MT attachment as previously though". They need to add "in the absence of active plk1"

Corrected.

9) Paragraph titles all along their results section would strongly help the reader.

Sub-headings are added.

10) It is unclear why the authors measured the efficiency of BLM depletion by siRNA as number of dots and not signal intensity measured by immuno-blot or RT-qPCR.

We now show both measurements of BLM nuclear intensity and an immuno-blot (New Supplementary Fig. 12b & Fig. 6a). It is worth noting that BLM expression is cell-cycle regulated (Dutertre et al., Oncogene, 2000), so G2/M cells exhibit higher nuclear signals, but after siBLM, there is a G2/M population showing reduced BLM nuclear signal, similar to that in G1 cells. We also provide representative images showing the lack of BLM (not PICH) foci at centromeres in the metaphase-like cells after BLM KD (New Supplementary Fig. 12g). The advantage of scoring BLM foci (and nuclear intensity) by ScanR is that we can measure the BLM depletion efficiency on the same sample preparation for DNA thread analysis. This provides an internal validation of the BLM knockdown. Scoring endogenous BLM foci numbers also minimises the background caused from non-specific nucleoplasmic signals. Nevertheless, we demonstrate that BLM depletion rescues centromere disintegration. This is also confirmed by using BLM knockout cells.

REVIEWERS' COMMENTS:

Reviewer #1 (Remarks to the Author):

The authors have significantly improved the manuscript and addressed all my concerns. I recommend publication.

Reviewer #2 (Remarks to the Author):

I feel that the authors have made a decent effort to address my (and the other referees' concerns). Although there are still a few puzzling aspects of the results, on balance I would now support publication.

Reviewer #3 (Remarks to the Author):

The revised version by Jones et al is a much more complete and improved version. They successfully replied to almost all my comments with a detailed explanation or new experiments (some successful, some not). I am in favor for publication in Nature Communications.

Few comments to consider to further improvement of their work:

1) The authors state in their rebuttal: "However, because of the asynchronicity, we notice that some mitotic cells that are already in late prometaphase and metaphase, when the drug is added, escape into anaphase. So, it is less efficient to see centromere disintegration from an asynchronous population compared to synchronous cells." I have found the argument of cells slipping through anaphase very confusing as the authors are looking at metaphase cells anyway. So, this anaphase cells should not influence their analysis. This means that thymidine treatment may worsen their phenotype. Maybe something to mention in the discussion.

2) I am not sure why they perform ICRF193+Plk1 inhibition at the same time. If they act in the same pathway, we're not expecting to see an effect. The idea was to see if they get the same kind of early centromeric DNA threads with ICRF or if Top2A levels/localization is affected by Plk1 inhibition.

3) They should show a quantification of the polo-like and fig-8-like phenotypes since they include that in their final model.

Reply to the reviewer's comments

We thank you again to all reviewers for their effort and time, especially for their helpful input to our study.

REVIEWERS' COMMENTS:

Reviewer #3 (Remarks to the Author):

The revised version by Jones et al is a much more complete and improved version. They successfully replied to almost all my comments with a detailed explanation or new experiments (some successful, some not). I am in favor for publication in Nature Communications.

We thank the reviewer for the support of our publication.

Few comments to consider to further improvement of their work:

1)The authors state in their rebuttal:"However, because of the asynchronicity, we notice that some mitotic cells that are already in late prometaphase and metaphase, when the drug is added, escape into anaphase. So, it is less efficient to see centromere disintegration from an asynchronous population compared to synchronous cells." I have found the argument of cells slipping through anaphase very confusing as the authors are looking at metaphase cells anyway. So, this anaphase cells should not influence their analysis. This means that thymidine treatment may worsen their phenotype. Maybe something to mention in the discussion.

We include the following sentence in the result section "....., thymidine pre-treatment or synchronistic mitotic entry may enhance the phenotype appearance."

2) I am not sure why they perform ICRF193+Plk1 inhibition at the same time. If they act in the same pathway, we're not expecting to see an effect. The idea was to see if they get the same kind of early centromeric DNA threads with ICRF or if Top2A levels/localization is affected by Plk1 inhibition.

The reason we treated cells simultaneously with TOP2 and PLK1 inhibitors was to test if TOP2A activity is required for PIT (centromeric DNA threads) formation that we interpreted from the reviewer's previous comment: "3) *Topoisomerase IIa has been extensively described to promote the resolution of centromeric DNA structures during mitosis (Porter and Farr, 2004), in association with PICH and BLM (Nielsen et al., 2015; Rouzeau et al., 2012). Given the importance of this protein in the processing of centromeric DNA, the authors have to show whether topo IIa activity is involved in PIT formation.*" We found that topoII α activity is not involved in PIT formation because the inhibition of TOP2 does not prevent PIT formation induced by BI2536.

We now understood the reviewer's point is to test if PIT formation may be caused by the loss of TOP2A activity or its mis-localisation (assuming because of PLK1 interference). First, we did not notice any obvious changes in TOP2A localisation on mitotic chromosomes (please see Fig. 2f). Second, extensive studies have been performed to examine the effects of TOP2A inhibition on mitotic chromosome organisation and mitotic

progression, and mostly, TOP2A inhibition does not lead to phenotypes similar to that of PLK1 inactivation. The loss of TOP2A causes under-condensation of chromosomes. However, PLK1-inhibited cells show more compact chromosomes. Besides, TOP2A inhibition, though delays mitosis, does not cause severe mitotic arrest, as cells still progress into anaphase but with massive DNA bridge formation. It also does not seem to cause any severe defects in KT-MT attachment or kinetochore disassembly. In fact, TOP2A inhibition reduces inter-kinetochore distances in metaphase(-like) cells, implying that centromere chromatin does not seem to get more loosened or disintegrated (Spence et al., 2007, JCS and Ribeiro et al., 2009, MBC). While we found the reviewer's suggestion is very interesting, given many well-known TOP2A phenotypes not resemble to PLK1 inactivation, we believe TOP2A (or its alone) is very unlikely to play a critical role in centromere disintegration. We also think the result will not change our key finding that the unlawful DNA unwinding at centromeres, together with spindle pulling forces, are the main driver of centromere disintegration. Therefore, we think this investigation should be left for future investigation.

3) They should show a quantification of the polo-like and fig-8-like phenotypes since they include that in their final model.

We thank the reviewer for the suggestion. Indeed, we prefer not to use the quantification of polo-like and fig-8-like misalignment phenotypes to determine centromere rupture. First, we think it is too subjective and indirect. Second, we do observe DNA thread formation (centromere damage) on mitotic cells exhibiting polo-like misalignment, especially in HAP1 and HCT116 cell lines. Thus, a polo-like phenotype does not necessarily mean there is no centromere rupture. A more objective and direct measurement, as we have used, is to quantify centromere DNA thread formation and centromere dislocation on mitotic spreads. We clearly show that depletion of PICH or BLM significantly suppresses centromere disintegration; but in general, siPICH/siBLM cells tend to display a polo-like misalignment pattern after prolonged PLK1 inhibition. Although we describe this interesting misalignment pattern, we do not include this quantification because as mentioned above, we think it is not a good indicator of centromere disintegration. Moreover, it is also worth to note that sister chromatids cohesion 'fatigue' can also generate a phenotype resemble to fig-8 misalignment. So, we do not think the quantification will provide additional important information. Finally, we have amended the legend of the model (Fig. 9), stating that whole chromosome misalignment probably manifests as more like a 'Polo' pattern.